# DISCRETE DIFFUSION TRAJECTORY ALIGNMENT VIA STEPWISE DECOMPOSITION

**Jiaqi Han**[1][†][*]     **Austin Wang**[2][†]     **Minkai Xu**[1]     **Wenda Chu**[2]

**Meihua Dang**[1]     **Haotian Ye**[1]     **Huayu Chen**[3]     **Yisong Yue**[2]     **Stefano Ermon**[1]

[1]Stanford University     [2]Caltech     [3]Tsinghua University

## ABSTRACT

Discrete diffusion models have demonstrated great promise in modeling various sequence data, ranging from human language to biological sequences. Inspired by the success of RL in language models, there is growing interest in further improving the models by alignment with a certain reward. In this work, we propose an offline preference optimization method to approach trajectory alignment for discrete diffusion models. Instead of applying the reward on the final output and backpropagating the gradient to the entire denoising process, we decompose the problem into a set of stepwise alignment objectives by matching the per-step posterior. This framework enables efficient diffusion optimization, is compatible with arbitrary reward functions, and importantly, yields an equivalent optimal solution under additive factorization of the trajectory reward. Experiments across multiple domains including DNA sequence design, protein inverse folding, and language modeling consistently demonstrate the superiority of our approach. Notably, it achieves an up to 12% improvement over the most competitive RL-based baseline in terms of predicted activity on DNA sequence design, and further improves the GSM8K score from 78.6 to 81.2 on LLaDA-8B-Instruct for language modeling.

## 1 INTRODUCTION

Diffusion models (DMs) (Sohl-Dickstein et al., 2015; Ho et al., 2020; Song et al., 2021) have emerged as a powerful tool for modeling distributions and generating samples across an array of modalities such as visual contents (Rombach et al., 2021; Saharia et al., 2022; Ho et al., 2022), natural languages (Nie et al., 2025; Lou et al., 2023; Shi et al., 2024; Sahoo et al., 2024), and geometric structures (Xu et al., 2022; Han et al., 2024b; Hoogeboom et al., 2022b), to name a few. Among them, discrete diffusion models (Austin et al., 2021; Campbell et al., 2022; Lou et al., 2023; Sahoo et al., 2024; Shi et al., 2024; Hoogeboom et al., 2022a), those that are in particular grounded on masked discrete latent variables, have demonstrated remarkable promise for modeling sequence data in discrete space, achieving superior performance on tasks ranging from DNA sequence design (Wang et al., 2024; Gosai et al., 2023) and protein inverse folding (Campbell et al., 2024; Wang et al., 2024; Hsu et al., 2022) to even text generation (Lou et al., 2023; Sahoo et al., 2024; Shi et al., 2024; Zheng et al., 2023; Gong et al., 2025) and chatbot (Nie et al., 2025; Ye et al., 2025).

Despite the promise, a critical question still remains unrevealed for discrete DMs: *How to align pretrained discrete diffusion models towards a target distribution, usually defined in the presence of certain reward?* Such problem has been of core interest in finetuning modern Large Language Models (LLMs) (Brown et al., 2020; Achiam et al., 2023; Team et al., 2023), a paradigm usually referred to as Reinforcement Learning with Human Feedback (RLHF) (Christiano et al., 2017; Stiennon et al., 2020; Ouyang et al., 2022) or preference optimization (Rafailov et al., 2023; Ji et al., 2024). It is vital in enhancing the applicability of the pretrained model on downstream tasks by biasing its distribution

---

[*]Correspondence to: jiaqihan@stanford.edu. Code is available at https://github.com/hanjq17/discrete-diffusion-sdpo.

[†]Equal contribution.

towards that with higher rewards, *e.g.*, higher enhancer activity for DNA sequence (Wang et al., 2024) or helpfulness and harmlessness for chatbots (Rafailov et al., 2023; Ji et al., 2024; Bai et al., 2022).

Existing alignment literature is primarily based on the left-to-right autoregressive modeling of sequences (Rafailov et al., 2023; Han et al., 2024a), and performing preference optimization is particularly challenging for discrete DMs, which hold the fundamentally different factorization with a Markov chain of sequence-level discrete random variables through a large number of diffusion steps. Previous work explored using RL to fine-tune the model, but the inherent discrete representation makes it challenging to efficiently backpropagate the gradient to the entire sampling process, with the reward typically computed upon the final output. Furthermore, this nature also makes it prohibitive to efficiently compute exact likelihood and evaluate rewards when aligning the joint of latent variables on the chain, leading to suboptimal performance (Wallace et al., 2024; Zhu et al., 2025b). The chained sampling of discrete diffusion also makes online RL (Zhao et al., 2025) computationally exhaustive.

In this work, we propose a principled approach for preference optimization of discrete diffusion models via *stepwise decomposition*. Our key innovation is to decompose the alignment of the entire diffusion trajectory $p_\theta(\mathbf{x}_{0:T})$ into a set of subproblems, each of which is responsible for aligning the per-step factorized approximation of the posterior $\hat{p}_\theta(\mathbf{x}_0|\mathbf{x}_t)$, where $\mathbf{x}_0$ is clean sequence and $\mathbf{x}_t$ is the latent variable at diffusion step $t$. Our stepwise decomposition takes the advantage of leveraging $\hat{p}_\theta(\mathbf{x}_0|\mathbf{x}_t)$ as the per-step alignment target, thus enabling both efficient and accurate likelihood computation and reward evaluation defined on clean sequence $\mathbf{x}_0$. Furthermore, we also theoretically reveal a novel connection between our stepwise decomposition alignment and the original problem by showing that the optimally aligned posteriors $\hat{p}^*(\mathbf{x}_0|\mathbf{x}_t)$ induce a joint $p^*(\mathbf{x}_{0:T})$ that is also an optimal solution of the diffusion trajectory alignment objective, when the reward of the trajectory takes an additive factorization over certain stepwise reward. In addition, we also develop a general form to align the stepwise posterior $\hat{p}_\theta(\mathbf{x}_0|\mathbf{x}_t)$ that works with arbitrary reward models, as opposed to previous preference optimization approaches (Rafailov et al., 2023; Wallace et al., 2024) specifically tailored under certain simplified reward such as the Bradley-Terry model (Bradley & Terry, 1952).

**Contributions.** To sum up, we propose stepwise decomposition preference optimization (SDPO) for offline finetuning of discrete diffusion models, with the following detailed contributions. **1.** We decompose the diffusion trajectory alignment problem into a set of subproblems that align the posterior $\hat{p}_\theta(\mathbf{x}_0|\mathbf{x}_t)$ for each diffusion step, allowing for efficient and exact likelihood and reward evaluation. **2.** We theoretically demonstrate the equivalence of SDPO and diffusion trajectory alignment through the bridge of certain stepwise reward. **3.** We derive a general loss function that jointly optimizes the stepwise alignment problems under arbitrary reward functions. **4.** We conduct extensive experimental evaluations on three different tasks, namely DNA sequence design, protein inverse folding, and language modeling. Our approach exhibits consistent enhancements, outperforming baselines by a significant margin across all benchmarks. Notably, we obtain a remarkable 12% gain in terms of predicted activity on the DNA sequence design, compared with the most competitive RL-based method (Wang et al., 2024; Borso et al., 2025) tailored for finetuning discrete diffusion models. Moreover, we adopt our approach to LLaDA-8B-Instruct (Nie et al., 2025), which further enhances GSM8K 5-shot score from 78.6 to 81.2, further demonstrating its promise as large language models.

## 2 RELATED WORK

**Discrete diffusion models.** Discrete diffusion models, originally formulated in Austin et al. (2021); Campbell et al. (2022); Hoogeboom et al. (2022a) and further extended by Lou et al. (2023); Sahoo et al. (2024); Shi et al. (2024); Zhao et al. (2024), have attracted growing interest in particular for modeling sequence data. Different from autoregressive models (Brown et al., 2020; Achiam et al., 2023; Team et al., 2023), discrete diffusion models relax from the inherent left-to-right causal ordering, allowing for more flexible modeling and parallel decoding (Xu et al., 2025; Zheng et al., 2025). They have achieved remarkable performance on various tasks, ranging from biological sequence design (Wang et al., 2024; Campbell et al., 2024) to human natural language modeling (Arriola et al., 2025; Nie et al., 2025; Ye et al., 2025; Zheng et al., 2023). Despite the promise, how to perform preference optimization on pretrained discrete diffusion models to align with certain reward still remains a challenge, which we aim to address in this work.

**Preference optimization for language models.** Aligning language models with certain reward is a core problem to enhance their utility (Ouyang et al., 2022). Initial approaches under the paradigm of

RLHF (Ouyang et al., 2022; Christiano et al., 2017) that employ RL-based algorithms (Schulman et al., 2017) for alignment have been proposed and successfully adopted. Direct preference optimization (DPO) (Rafailov et al., 2023) and subsequent works (Ethayarajh et al., 2024; Meng et al., 2024; Han et al., 2024a; Ji et al., 2024; Lai et al., 2024) leverage pairwise or ranking-based preference dataset to perform offline optimization that further address the optimization instability and complexity. Whilst much progress have been made, they are developed upon autoregressive language models, while we instead focus on discrete diffusion models with a substantially different probabilistic factorization.

**Diffusion alignment.** Preference optimization has also been explored for diffusion models. The pioneer attempt of Wallace et al. (2024); Yang et al. (2024) extend DPO to Gaussian diffusion and is able to promote image quality. Li et al. (2024); Gu et al. (2024) further improve the performance by employing different human preference modeling while Zhu et al. (2025b) proposes to align the score function. There are also works that resort to RL (Fan et al., 2023; Black et al., 2024) or directly backpropagating through differentiable reward (Clark et al., 2024; Prabhudesai et al., 2024). Differently, we develop a principled objective for discrete diffusion which pose unique challenges due to the discrete nature. Wang et al. (2024) approaches this problem through RL by backpropagating the gradient via the Gumbel trick, which leads to optimization overhead. Recent works also derive under pairwise preference based on DPO (Borso et al., 2025; Zhu et al., 2025a), resort to online sampling and verification (Zhao et al., 2025; Yang et al., 2025), or perform optimization through posterior matching (Rector-Brooks et al., 2025). Besides these training-based approaches, inference-time guidance has also been explored for discrete diffusion to align sampling distributions. Nisonoff et al. (2024) adapts classifier-guidance (Ho & Salimans, 2022) to discrete diffusion, while sequential Monte Carlo (SMC)-based approaches (Wu et al., 2023; Phillips et al., 2024; Dou & Song, 2024) have also been introduced for more effective guidance. However, these guidance-based methods usually induce much higher sampling cost and easily suffer from suboptimal performance when the guidance signal is insufficient. Critically, our approach instead offers a generalized optimization objective, does not require online sampling at each iteration, and demonstrates enhanced performance.

## 3 PRELIMINARIES

**Discrete diffusion models.** Discrete diffusion models (Austin et al., 2021; Lou et al., 2023; Sahoo et al., 2024; Shi et al., 2024) are a family of diffusion models with the latent variables residing in the discrete space $\mathcal{X}$ with dimensionality $m$. With input data point $\mathbf{x}_0$, discrete diffusion features a forward diffusion process in the form of Markov chain $q(\mathbf{x}_t|\mathbf{x}_0)$ with

$$q(\mathbf{x}_t|\mathbf{x}_0) := \mathrm{Cat}\left(\mathbf{x}_t; \alpha_t \mathbf{x}_0 + (1 - \alpha_t)\boldsymbol{\pi}\right), \tag{1}$$

where $\boldsymbol{\pi}$ is the vectorized representation of certain prior distribution $\mathrm{Cat}\left(\cdot; \boldsymbol{\pi}\right)$, and $\alpha_t$, usually referred to as the noise schedule, is a decreasing function *w.r.t.* $t$ satisfying that $\alpha_0 = 1$ and $\alpha_T = 0$. The transition for any two timesteps $0 \leq s \leq t \leq T$ that induces $q(\mathbf{x}_t|\mathbf{x}_0)$ is specified as $q(\mathbf{x}_t|\mathbf{x}_s) = \mathrm{Cat}\left(\mathbf{x}_t; \alpha_{t|s}\mathbf{x}_s + (1 - \alpha_{t|s})\boldsymbol{\pi}\right)$ where $\alpha_{t|s} = \alpha_t/\alpha_s$.

**Masked discrete diffusion models.** Masked discrete diffusion models (Sahoo et al., 2024; Shi et al., 2024; Lou et al., 2023; Austin et al., 2021) are discrete diffusion models when the prior $\boldsymbol{\pi}$ is in particular instantiated as the absorbing state $\mathbf{m} := [0, \cdots, 0, 1]$ where the last entry in $\mathbf{m}$ corresponds to a special MASK token. The posterior has a simplified form (Sahoo et al., 2024; Shi et al., 2024):

$$q(\mathbf{x}_s|\mathbf{x}_t, \mathbf{x}_0) = \begin{cases} \mathrm{Cat}(\mathbf{x}_s; \mathbf{x}_t) & \mathbf{x}_t \neq \mathbf{m}, \\ \mathrm{Cat}\left(\mathbf{x}_s; \frac{\alpha_s - \alpha_t}{1 - \alpha_t}\mathbf{x}_0 + \frac{1 - \alpha_s}{1 - \alpha_t}\mathbf{m}\right) & \mathbf{x}_t = \mathbf{m}. \end{cases} \tag{2}$$

The reversal $p_\theta(\mathbf{x}_s|\mathbf{x}_t)$ is then parameterized by a neural network $\mathbf{f}_\theta(\mathbf{x}_t, t)$ that predicts $\mathbf{x}_0$ in Eq. 2, which is optimized to approximate the posterior by minimizing the negative evidence lower bound $-\log p(\mathbf{x}_0) \leq \mathcal{L}_{\mathrm{NELBO}} := \mathbb{E}_{q(\mathbf{x}_t|\mathbf{x}_0)} \sum_{t=1}^{t=T} \frac{\alpha_t - \alpha_{t-1}}{1 - \alpha_t} \log\left(\mathbf{x}_0^\top \cdot \mathbf{f}_\theta(\mathbf{x}_t, t)\right)$.

**Reinforcement learning with human feedback.** At alignment stage, a pretrained model $p_\theta(\mathbf{x}|\mathbf{c})$ is finetuned to maximize certain reward $r(\mathbf{x}, \mathbf{c})$ subject to a Kullback–Leibler (KL) divergence regularization *w.r.t.* the reference model $p_{\mathrm{ref}}(\mathbf{x}|\mathbf{c})$, leading to the following objective:

$$\max_{p_\theta} \mathbb{E}_{\mathbf{x}, \mathbf{c}}\left[r(\mathbf{x}, \mathbf{c})\right] - \beta D_{\mathrm{KL}}\left[p_\theta(\mathbf{x}|\mathbf{c})\|p_{\mathrm{ref}}(\mathbf{x}|\mathbf{c})\right], \tag{3}$$

where $\mathbf{c}$ is some context such as a prompt and $\beta$ is the balancing factor. The choice of the reward model can be arbitrary, such as human or LLM-assisted preference labels (Ouyang et al., 2022; Rafailov et al., 2023), or the predicted activity of the designed DNA sequence (Wang et al., 2024). This KL-constrained optimization problem has the optimal solution (Peters & Schaal, 2007)

$$p^*(\mathbf{x}|\mathbf{c}) = \frac{1}{Z(\mathbf{c})} p_{\text{ref}}(\mathbf{x}|\mathbf{c}) \exp\left(\frac{1}{\beta} r(\mathbf{x}, \mathbf{c})\right), \tag{4}$$

where $Z(\mathbf{c}) = \sum_{\mathbf{x}} p_{\text{ref}}(\mathbf{x}|\mathbf{c}) \exp\left(\frac{1}{\beta} r(\mathbf{x}, \mathbf{c})\right)$ is the partition function that is intractable to evaluate.

**Problem formulation.** In this work, we aim to develop an efficient offline alignment approach for discrete diffusion models. Specifically, the algorithm directly operates on a pre-collected dataset $\mathcal{D} = \{(\mathbf{x}_0, \mathbf{c}, r(\mathbf{x}_0, \mathbf{c}))\}$ on clean data $\mathbf{x}_0$ without relying on on-policy generations during finetuning.

## 4 METHOD

In this section, we detail our approach for aligning discrete diffusion models through stepwise optimization. In § 4.1, we first revisit the problem of discrete diffusion alignment and investigate the challenges. In § 4.2, we propose a novel stepwise decomposition approach for discrete diffusion alignment. In § 4.3, we introduce a principled way to solve the stepwise alignment objective through distribution matching. We offer additional in-depth analyses and discussions in § 4.4.

### 4.1 ALIGNING DISCRETE DIFFUSION MODELS

Different from autoregressive models that can evaluate $p_\theta(\mathbf{x}|\mathbf{c})$ efficiently in a single forward pass, discrete diffusion models are grounded on a chain of random variables $\mathbf{x}_{0:T} := [\mathbf{x}_0, \mathbf{x}_1, \cdots, \mathbf{x}_T]$, where the joint satisfies the Markovian factorization $p_\theta(\mathbf{x}_{0:T}|\mathbf{c}) = p_\theta(\mathbf{x}_T|\mathbf{c}) \prod_{t=1}^{T} p_\theta(\mathbf{x}_{t-1}|\mathbf{x}_t, \mathbf{c})$. The alignment objective in Eq. 3 is therefore extended to the entire chain (Wallace et al., 2024):

$$\max_{p_\theta} \mathbb{E}_{p_\theta(\mathbf{x}_{0:T}|\mathbf{c}),\mathbf{c}} \left[\hat{r}(\mathbf{x}_{0:T}, \mathbf{c})\right] - \beta D_{\text{KL}} \left[p_\theta(\mathbf{x}_{0:T}|\mathbf{c}) \| p_{\text{ref}}(\mathbf{x}_{0:T}|\mathbf{c})\right], \tag{5}$$

where the reward $\hat{r}(\mathbf{x}_{0:T}, \mathbf{c})$ now considers the whole chain $\mathbf{x}_{0:T}$. We hence refer to the optimization problem of Eq. 5 as *diffusion trajectory optimization*. Akin to Eq. 4, the optimal solution is $p^*(\mathbf{x}_{0:T}|\mathbf{c}) = \frac{1}{Z(\mathbf{c})} p_{\text{ref}}(\mathbf{x}_{0:T}|\mathbf{c}) \exp\left(\frac{1}{\beta} \hat{r}(\mathbf{x}_{0:T}, \mathbf{c})\right)$. However, the optimization problem in Eq. 5 poses several challenges. First, the expectation is taken over the entire chain $p_\theta(\mathbf{x}_{0:T})$, making it computationally expensive. Moreover, the reward $\hat{r}(\mathbf{x}_{0:T}, \mathbf{c})$ is supposed to operate on the entire chain, while empirical rewards $r(\mathbf{x}_0, \mathbf{c})$, *e.g.*, human preference (Rafailov et al., 2023) or DNA activity (Wang et al., 2024), are most commonly defined on the clean sequence $\mathbf{x}_0$. We will introduce our stepwise decomposition approach that offers a simplified and tractable measure to solve Eq. 5.

### 4.2 DIFFUSION TRAJECTORY OPTIMIZATION THROUGH STEPWISE DECOMPOSITION

We propose a principled way to solve the problem by decomposing the trajectory optimization into a set of subproblems, each of which corresponds to a *stepwise* alignment objective for the factorized approximation of the posterior $\hat{p}_\theta(\mathbf{x}_0|\mathbf{x}_t, \mathbf{c}) := \prod_{i=1}^{L} \hat{p}_\theta(\mathbf{x}_0^{(i)}|\mathbf{x}_t, \mathbf{c})$ (Shi et al., 2024; Austin et al., 2021) at diffusion step $1 \leq t \leq T$ (see Fig. 1), where $i$ is the token index and $L$ is the sequence length. To be specific, the set of subproblems is

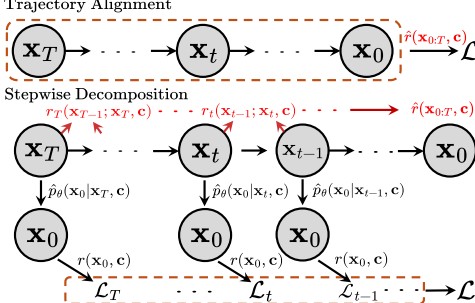

Figure 1: The flowchart of our SDPO.

$$\max_{\hat{p}_\theta} \mathbb{E}_{\hat{p}_\theta(\mathbf{x}_0|\mathbf{x}_t, \mathbf{c}),\mathbf{c}} \left[r(\mathbf{x}_0, \mathbf{c})\right]$$
$$- \beta_t D_{\text{KL}} \left[\hat{p}_\theta(\mathbf{x}_0|\mathbf{x}_t, \mathbf{c}) \| \hat{p}_{\text{ref}}(\mathbf{x}_0|\mathbf{x}_t, \mathbf{c})\right], \tag{6}$$

$\forall 1 \leq t \leq T$, where $\beta_t = \beta/w(t)$ is the stepwise regularization reweighted by certain scheduler $w(t)$. The optimal solution for any $t$ is given by $\hat{p}^*(\mathbf{x}_0|\mathbf{x}_t, \mathbf{c}) = \frac{1}{Z(\mathbf{c})} \hat{p}_{\text{ref}}(\mathbf{x}_0|\mathbf{x}_t, \mathbf{c}) \exp\left(\frac{1}{\beta_t} r(\mathbf{x}_0, \mathbf{c})\right)$.

Such a formulation enjoys several unique benefits compared with the trajectory alignment objective in Eq. 5. First, the expectation over the entire chain $p_\theta(\mathbf{x}_{0:T}|\mathbf{c})$ has been decomposed into the stepwise posterior $\hat{p}_\theta(\mathbf{x}_0|\mathbf{x}_t, \mathbf{c})$, which can be computed both tractably and efficiently for discrete diffusion models. Furthermore, by grounding on the clean data $\mathbf{x}_0$ instead of intermediate latent variables $\mathbf{x}_t$, we can readily reuse the reward model $r(\mathbf{x}_0, \mathbf{c})$ without resorting to its biased estimates (Lu et al., 2023; Chen et al., 2024). More interestingly, below we reveal a critical connection between the stepwise decomposition alignment objective (Eq. 6) and the trajectory optimization objective (Eq. 5), as stated in the theorem below:

**Theorem 4.1.** *The joint $p^*(\mathbf{x}_{0:T}|\mathbf{c})$ induced by the optimal solutions $\{\hat{p}^*(\mathbf{x}_0|\mathbf{x}_t, \mathbf{c})\}_{t=1}^T$ of Eq. 6 is also the optimal solution of the trajectory alignment objective in Eq. 5, with the chain reward*

$$\hat{r}(\mathbf{x}_{0:T}, \mathbf{c}) = \beta \sum_{t=1}^T r_t(\mathbf{x}_{t-1}; \mathbf{x}_t, \mathbf{c}) \ where \ r_t(\mathbf{x}_{t-1}; \mathbf{x}_t, \mathbf{c}) = \log \frac{\mathbb{E}_{p'_{\mathrm{ref}}(\mathbf{x}_0|\mathbf{x}_{t-1}, \mathbf{x}_t, \mathbf{c})}\left[\exp\left(\frac{1}{\beta_t} r(\mathbf{x}_0, \mathbf{c})\right)\right]}{\mathbb{E}_{p_{\mathrm{ref}}(\mathbf{x}_0|\mathbf{x}_t, \mathbf{c})}\left[\exp\left(\frac{1}{\beta_t} r(\mathbf{x}_0, \mathbf{c})\right)\right]}.$$

Proof is in Appendix A.1. Here $p'_{\mathrm{ref}}(\mathbf{x}_0|\mathbf{x}_{t-1}, \mathbf{x}_t, \mathbf{c}) := \frac{\hat{p}_{\mathrm{ref}}(\mathbf{x}_0|\mathbf{x}_t)q(\mathbf{x}_{t-1}|\mathbf{x}_0, \mathbf{x}_t)}{p_{\mathrm{ref}}(\mathbf{x}_{t-1}|\mathbf{x}_t)}$ is the posterior of $\mathbf{x}_0$ *w.r.t.* a specific choice of $\mathbf{x}_{t-1}$, given $\mathbf{x}_t$. In the case of masked diffusion models, the posterior refers to the factorized conditional $\hat{p}_{\mathrm{ref}}(\mathbf{x}_0|\mathbf{x}_t, \mathbf{c})$ constrained on the set of all possible $\mathbf{x}_0$ that share the same decoded tokens with $\mathbf{x}_{t-1}$. Theorem 4.1 endorses our key finding that the intractable trajectory optimization can be alternatively approached by jointly optimizing the *stepwise* alignment objectives, under which the reward of the chain $\hat{r}(\mathbf{x}_{0:T}, \mathbf{c})$ is effectively an additive factorization of the *stepwise reward* $r_t(\mathbf{x}_{t-1}; \mathbf{x}_t, \mathbf{c})$. More interestingly, the stepwise reward also has intuitive implications. At each diffusion step $t$ with the sampled $\mathbf{x}_t$, the denominator inside $\log$ is a constant and $r_t$ is therefore distinguished fully by the numerator, a term that effectively assigns higher reward to those $\mathbf{x}_{t-1}$ who are more likely to be obtained from the $\mathbf{x}_0$ with higher reward $r(\mathbf{x}_0, \mathbf{c})$. Furthermore, the stepwise rewards also serve as more fine-grained supervision that enables tractable alignment of each diffusion step, while previous works that operate fully on the trajectory-level confer no per-step guarantee.

## 4.3 GENERALIZED STEPWISE ALIGNMENT THROUGH DISTRIBUTION MATCHING

While the stepwise decomposition has introduced clear benefits, it is still yet unclear how to optimize $\hat{p}_\theta(\mathbf{x}_0|\mathbf{x}_t, \mathbf{c})$ towards the optimal solution $\hat{p}^*(\mathbf{x}_0|\mathbf{x}_t, \mathbf{c})$, particularly under arbitrary reward $r(\mathbf{x}_0, \mathbf{c})$. To this end, existing works seek to directly backpropagate the gradient from the reward model (Wang et al., 2024), which inevitably incurs optimization overhead and instability, or to simplify the reward into tractable forms such as the Bradley-Terry model (Wallace et al., 2024), which imposes additional constraints. Differently, we propose to perform optimization based on the following objective:

$$\mathcal{L}_t(\theta) := \mathbb{E}_{\mathbf{x}_t, \mathbf{c}}\left[D_{\mathrm{KL}}\left[\tilde{p}_r(\mathbf{x}_0|\mathbf{x}_t, \mathbf{c})\|\tilde{p}_\theta(\mathbf{x}_0|\mathbf{x}_t, \mathbf{c})\right]\right], \tag{7}$$

where $\tilde{p}_r(\mathbf{x}_0|\mathbf{x}_t, \mathbf{c}) \propto \hat{p}_{\mathrm{ref}}(\mathbf{x}_0|\mathbf{x}_t, \mathbf{c}) \exp(r(\mathbf{x}_0, \mathbf{c}))$ is the Boltzmann policy (Laidlaw & Dragan, 2022; Peters & Schaal, 2007) induced by the reward then reweighted by $p_{\mathrm{ref}}$, while $\tilde{p}_\theta(\mathbf{x}_0|\mathbf{x}_t, \mathbf{c}) \propto \hat{p}_{\mathrm{ref}}(\mathbf{x}_0|\mathbf{x}_t, \mathbf{c})^{(1-\beta_t)}\hat{p}_\theta(\mathbf{x}_0|\mathbf{x}_t, \mathbf{c})^{\beta_t}$ is similarly the reweighted model policy. The rationale of Eq. 7 lies in that the minimizer of this KL-divergence distribution matching (Han et al., 2024a; Ji et al., 2024) problem is also the optimal solution of stepwise alignment (proof in Appendix. A.2):

**Proposition 4.2.** *Let $\theta^* = \arg\min \mathcal{L}_t(\theta)$ defined in Eq. 7. Then $\hat{p}_{\theta^*}(\mathbf{x}_0|\mathbf{x}_t, \mathbf{c}) = \hat{p}^*(\mathbf{x}_0|\mathbf{x}_t, \mathbf{c})$, the optimal solution of the stepwise alignment objective in Eq. 6.*

Besides the guaranteed equivalence of the optimal solution, the definition of $\tilde{p}_r$ also enables importance sampling by using $p_{\mathrm{ref}}$ as the proposal distribution, from which the offline preference datasets are drawn. Expanding Eq. 7 with importance sampling (see Appendix. A.3), we have

$$\mathcal{L}_t(\theta) = -\mathbb{E}_{\mathbf{c}, \hat{p}_{\mathrm{ref}}(\mathbf{x}_0, \mathbf{x}_t|\mathbf{c})}\left[\frac{\exp(r(\mathbf{x}_0, \mathbf{c}))}{Z_r(\mathbf{c})} \log \frac{\exp(r_\theta(\mathbf{x}_0, \mathbf{x}_t, \mathbf{c}, \beta_t))}{Z_\theta^t(\mathbf{x}_t, \mathbf{c}, \beta_t)}\right] + C, \tag{8}$$

where $r_\theta(\mathbf{x}_0, \mathbf{x}_t, \mathbf{c}, \beta_t) = \beta_t\left(\log \hat{p}_\theta(\mathbf{x}_0|\mathbf{x}_t, \mathbf{c}) - \log \hat{p}_{\mathrm{ref}}(\mathbf{x}_0|\mathbf{x}_t, \mathbf{c})\right)$ refers to the implicit reward (Rafailov et al., 2023; Cui et al., 2025), $Z_r(\mathbf{c}) = \mathbb{E}_{p_{\mathrm{ref}}(\mathbf{x}_0|\mathbf{c})} \exp(r(\mathbf{x}_0, \mathbf{c}))$ and $Z_\theta^t(\mathbf{x}_t, \mathbf{c}, \beta_t) = \mathbb{E}_{\hat{p}_{\mathrm{ref}}(\mathbf{x}_0|\mathbf{x}_t, \mathbf{c})} \exp(r_\theta(\mathbf{x}_0, \mathbf{x}_t^{(i)}, \mathbf{c}, \beta_t))$ are the partition functions, and $C$ is a constant irrelevant to $\theta$.

**Empirical form.** We leverage Monte-Carlo to estimate $\mathcal{L}_t$ as well as the partitions using $N$ samples $\{(\mathbf{x}_0^{(i)}, \mathbf{x}_t^{(i)}, \mathbf{c})\}_{i=1}^N$ drawn from $p_{\mathrm{ref}}(\mathbf{x}_0, \mathbf{x}_t|\mathbf{c})$ for each $\mathbf{c}$. In form, we employ

$$\tilde{\mathcal{L}}_t^N(\theta) = -\mathbb{E}_{\mathbf{c}} \sum_{i=1}^N \left(\frac{\exp(r(\mathbf{x}_0^{(i)}, \mathbf{c}))}{\sum_{j=1}^N \exp(r(\mathbf{x}_0^{(j)}, \mathbf{c}))} \cdot \log \frac{\exp(\tilde{r}_\theta(\mathbf{x}_0^{(i)}, \mathbf{x}_t^{(i)}, \mathbf{c}, \beta_t))}{\sum_{j=1}^N \exp(\tilde{r}_\theta(\mathbf{x}_0^{(j)}, \mathbf{x}_t^{(j)}, \mathbf{c}, \beta_t))}\right). \tag{9}$$

Eq. 9 takes the form of cross-entropy loss (Ji et al., 2024; Lu et al., 2023) between the self-normalized Boltzmann policies induced by $r(\mathbf{x}_0, \mathbf{c})$ and $\tilde{r}_\theta(\mathbf{x}_0, \mathbf{x}_t, \mathbf{c}, \beta_t)$. As $N \to \infty$, the estimate for the policy of $r$ becomes unbiased, while an unbiased estimate of $Z_\theta^t$ requires extensive sampling from the posterior $\hat{p}_{\text{ref}}(\mathbf{x}_0|\mathbf{x}_t, \mathbf{c})$ for each $\mathbf{x}_t$, which is highly prohibitive in the *offline* alignment setup. In practice we still favor the simplified MC estimate in Eq. 9 which is efficient and performant. We henceforth employ $\tilde{\mathcal{L}}_t^N(\theta)$ to solve each subproblem of Eq. 6. For the sample size $N$, we view it as a hyperparameter that trades off between efficiency and bias, depending on the task and dataset.

## 4.4 Overall Objective

Since the final objective (Eq. 6) requires to jointly optimize for the subproblems across all diffusion steps, at each iteration we randomly select a batch of diffusion steps, and optimize the corresponding $\mathcal{L}_t^N$ as per Eq. 9. Furthermore, since in offline settings the intermediate samples $\mathbf{x}_t$ are not preserved, we instead keep track of the clean samples $\mathbf{x}_0$ obtained from $p_{\text{ref}}$ while approaching the corresponding $\mathbf{x}_t$ via the forward process $q(\mathbf{x}_t|\mathbf{x}_0)$ at each training step. Putting all together we obtain our final loss

$$\mathcal{L}(\theta) = -\mathbb{E}_{t,\mathbf{c},\mathbf{x}_0,q(\mathbf{x}_t|\mathbf{x}_0)} \sum_{i=1}^{N} \left( \frac{\exp(r(\mathbf{x}_0^{(i)}, \mathbf{c}))}{\sum_{j=1}^{N} \exp(r(\mathbf{x}_0^{(j)}, \mathbf{c}))} \cdot \log \frac{\exp(\tilde{r}_\theta(\mathbf{x}_0^{(i)}, \mathbf{x}_t^{(i)}, \mathbf{c}, \beta_t))}{\sum_{j=1}^{N} \exp(\tilde{r}_\theta(\mathbf{x}_0^{(j)}, \mathbf{x}_t^{(j)}, \mathbf{c}, \beta_t))} \right),$$
(10)

where $\tilde{r}_\theta$, by further leveraging the reversal factorized parameterization of masked diffusion models (Shi et al., 2024; Sahoo et al., 2024) and the definition $\beta_t = \beta/w(t)$, takes the following form:

$$\tilde{r}_\theta(\mathbf{x}_0, \mathbf{x}_t, \mathbf{c}, \beta_t) = \beta \left( \frac{\log(\mathbf{x}_0^\top \mathbf{f}_\theta(\mathbf{x}_t, t, \mathbf{c}))}{w(t)} - \frac{\log(\mathbf{x}_0^\top \mathbf{f}_{\text{ref}}(\mathbf{x}_t, t, \mathbf{c}))}{w(t)} \right).$$
(11)

We note that our method applies to general discrete diffusion, but we choose to focus specifically on the masked variant. Our final loss has several implications, which we will analyze below.

**Pairwise preference data.** Our loss possesses a generalized form *w.r.t.* the reward model $r(\mathbf{x}_0, \mathbf{c})$ and $N$, *i.e.*, the number of samples for each context or prompt $\mathbf{c}$. In particular, it subsumes the setting in DPO where each prompt is provided with a pair of winning and losing completions $(\mathbf{x}_0^{(w)}, \mathbf{x}_0^{(l)})$, by setting $N = 2$ and leveraging Bradley-Terry (BT) model as the reward, *i.e.*, $r(\mathbf{x}_0^{(w)}, \mathbf{c}) = 0$ and $r(\mathbf{x}_0^{(l)}, \mathbf{c}) = -\infty$. We provide detailed derivations of our loss in this special case in Appendix A.4.

**The role of $w(t)$.** The coefficient $w(t)$ is initially introduced as the weight for the per-step reward $\hat{r}_t$. Interestingly, from Eq. 11 we can also interpret $w(t)$ as a factor that controls the scale of $\log(\mathbf{x}_0^\top \mathbf{f}(\mathbf{x}_t, t))$, which is correlated to the number of masked tokens at step $t$. Therefore we set $w(t) = 1 - \alpha_t$ to amortize the loss to each token, and empirically find this choice effective.

**The role of $\beta$.** Eq. 5 reveals that $\beta$ controls the strength of the KL regularization *w.r.t.* the reference distribution, which is also widely reflected in literature (Rafailov et al., 2023; Wallace et al., 2024).

**Iterative labeling.** Empirically we have also explored a variant of our approach that updates the dataset with samples from the latest model and their corresponding rewards. We find this iterative labeling generally favorable since more useful rewards are progressively provided for samples of higher quality, as the training proceeds. We defer detailed justifications to § 5.4.

## 5 Experiments

We perform extensive evaluations on a wide suite of tasks, including DNA sequence design (§ 5.1), protein inverse folding (§ 5.2), and language modeling (§. 5.3). We provide ablation studies in § 5.4.

## 5.1 DNA Sequence Design

We aim to finetune our model to unconditionally generate DNA sequences that trigger gene expression in targeted cell types. This is a task commonly seen in cell and gene therapy (Taskiran et al., 2024).

**Experiment setup.** We use a publicly available dataset (Gosai et al., 2023) that contains the measured enhancer activity in $\sim 700k$ DNA sequences, each 200 base-pairs in length. Cell line activity is

Table 1: Model performance on DNA sequence design. Our approach generates sequences with high activity while being natural-like. Results averaged across 3 random seeds with standard deviations in parentheses. Numbers of baselines are taken from Wang et al. (2024).

| | Pred-Activity (med) ↑ | ATAC-Acc ↑ (%) | 3-mer Corr ↑ | JASPAR Corr ↑ | App-Log-Lik (med) ↑ | Entropy (med) ↑ |
|---|---|---|---|---|---|---|
| Pretrained (Sahoo et al., 2024) | 0.17 (0.04) | 1.5 (0.2) | -0.061 (0.034) | 0.249 (0.015) | -261 (0.6) | **390** (6.2) |
| CG (Nisonoff et al., 2024) | 3.30 (0.00) | 0.0 (0.0) | -0.065 (0.001) | 0.212 (0.035) | -266 (0.6) | 12 (4.1) |
| SMC (Wu et al., 2023) | 4.15 (0.33) | 39.9 (8.7) | 0.840 (0.045) | 0.756 (0.068) | -259 (2.5) | 351 (6.5) |
| TDS (Wu et al., 2023) | 4.64 (0.21) | 45.3 (16.4) | 0.848 (0.008) | 0.846 (0.044) | -257 (1.5) | 340 (5.4) |
| CFG (Ho & Salimans, 2022) | 5.04 (0.06) | 92.1 (0.9) | 0.746 (0.001) | 0.864 (0.011) | -265 (0.6) | 363 (6.1) |
| D2-DPO (Borso et al., 2025) | 2.97 (0.03) | 35.6 (0.9) | **0.944** (0.002) | 0.883 (0.005) | -252 (0.4) | 362 (4.9) |
| VRPO (Zhu et al., 2025a) | 4.60 (0.01) | 15.8 (0.2) | 0.838 (0.005) | 0.865 (0.005) | -255 (0.8) | 289 (13.5) |
| DDPP-IS (Rector-Brooks et al., 2025) | 4.07 (0.02) | 50.0 (0.3) | 0.711 (0.001) | 0.723 (0.004) | -253 (0.9) | 378 (5.8) |
| DRAKES (Wang et al., 2024) | 5.61 (0.07) | 92.5 (0.6) | 0.887 (0.002) | 0.911 (0.002) | -264 (0.6) | 375 (5.2) |
| diffu-GRPO (Zhao et al., 2025) | 5.86 (0.04) | 33.0 (0.8) | 0.783 (0.001) | 0.903 (0.004) | **-245** (0.4) | 310 (8.6) |
| SDPO | **6.30** (0.003) | **94.8** (0.01) | 0.900 (0.003) | **0.936** (0.003) | -246 (0.5) | 365 (4.4) |

measured for each sequence, quantified with massively parallel reporter assays (MPRAs) that record the expression each sequence drives. The pre-trained masked diffusion language model (Sahoo et al., 2024) is taken from Wang et al. (2024), trained on the entire enhancer dataset. The pre-trained finetuning and evaluation reward models predict the HepG2 cell line activity in a sequence, also taken from Wang et al. (2024) and trained on different splits of the dataset.

**Baselines.** We compare with the following baselines. *Pretrained*: the base pre-trained model (no finetuning). *Guidance methods*: classifier guidance (*CG*) (Nisonoff et al., 2024), classifier-free guidance (CFG) (Ho & Salimans, 2022) and two Sequential Monte Carlo-based methods (Wu et al., 2023), namely *SMC*, where the proposal is the pretrained model, and *TDS*, where the proposal is *CG*. *D2-DPO* (Borso et al., 2025) and *VRPO* (Zhu et al., 2025a): offline preference optimization algorithms that adapt DPO to discrete diffusion. *DRAKES* (Wang et al., 2024): an online RL algorithm that backpropagates the reward through the generated trajectory with Gumbel-Softmax. diffu-GRPO (Zhao et al., 2025): a policy gradient-based approach for discrete diffusion. DDPP-IS (Rector-Brooks et al., 2025): an importance sampling method to match the reward-tilted posterior.

**Metrics.** We use the metrics following the protocol in Wang et al. (2024) to evaluate the model's enhancer generation. **1.** *Pred-Activity.* The enhancer activity level in the HepG2 cell line is predicted by the evaluation reward model, trained on a held out evaluation set. **2.** *ATAC-Acc.* We measure the proportion of generated sequences with high chromatin accessibility. This metric is typically correlated with the enhancer activity. **3.** *3-mer Corr.* We compute the 3-mer Pearson correlation between the generated sequences and the sequences from the enhancer dataset with the top 0.1% HepG2 activity. More natural, in-distribution sequences tend to have higher 3-mer Pearson correlation values. **4.** *JASPAR-Corr.* We compute potential transcription factor binding motifs in the generated sequences with JASPAR transcription factor binding profiles (Castro-Mondragon et al., 2022), and calculate the Spearman correlation of motif frequency between the generated samples and the top 0.1% sequences in the dataset with the highest activity. **5.** *App-Log-Lik.* The approximated log-likelihood of the generated sequences is computed with respect to the pre-trained model using the discrete diffusion ELBO presented in Sahoo et al. (2024). This metric evaluates the naturalness of the generations, as samples that over-optimize for the reward model tend to have worse log-likelihoods. **6.** *Entropy.* Sequence entropy is computed following Wang et al. (2024) to measure the sample diversity.

**Results.** Our method generates sequences that are both natural-like and have high predicted enhancer activity. Notably, we are able to significantly outperform all previous baselines in the predicted HepG2 activity, while also achieving strong 3-mer Pearson and JASPAR correlation numbers, demonstrating our method's robustness to over-optimizing for the reward model. In particular, we outperform the RL-based approach DRAKES by a significant margin of 12.3% in terms of predicted activity. The ATAC accuracy, another metric correlated with HepG2 activity, provides further validation of the high quality of our generated samples, as we see that other baselines, such as the SMC-based methods, may achieve relatively higher predicted enhancer activity but suffer poor ATAC accuracy numbers.

**Training efficiency.** Besides the superior performance, another feature that worth highlighting for SDPO is its training efficiency, since it does not require on-policy sampling at each training iteration. We report the average wallclock time per training step, where DRAKES takes 6.02 sec, diffu-GRPO takes 1.51 sec, and SDPO only takes 0.77 sec, which verifies the superior training efficiency of SDPO.

## 5.2 PROTEIN INVERSE FOLDING

For the protein inverse folding task, we finetune a pre-trained model that predicts the protein sequence from a 3D structure. We aim to optimize the stability of the protein sequences.

Table 2: Model performance on inverse protein folding. Our approach generates protein sequences with high stability and desired structure. Results averaged across 3 random seeds with standard deviations in parentheses. Numbers of baselines are taken from Wang et al. (2024).

| | Pred-ddG (med)↑ | %(ddG> 0) (%)↑ | scRMSD (med)↓ | %(scRMSD< 2)(%)↑ | Success Rate (%)↑ | Entropy (med)↑ |
|---|---|---|---|---|---|---|
| Pretrained (Campbell et al., 2024) | -0.544 (0.037) | 36.6 (1.0) | 0.849 (0.013) | 90.9 (0.6) | 34.4 (0.5) | 35.2 (8.1) |
| CG (Nisonoff et al., 2024) | -0.561 (0.045) | 36.9 (1.1) | 0.839 (0.012) | 90.9 (0.6) | 34.7 (0.9) | 34.6 (7.1) |
| SMC (Wu et al., 2023) | 0.659 (0.044) | 68.5 (3.1) | 0.841 (0.006) | 93.8 (0.4) | 63.6 (4.0) | 24.9 (6.9) |
| TDS (Wu et al., 2023) | 0.674 (0.086) | 68.2 (2.4) | 0.834 (0.001) | 94.4 (1.2) | 62.9 (2.8) | 24.9 (7.2) |
| CFG (Ho & Salimans, 2022) | -1.186 (0.035) | 11.0 (0.4) | 3.146 (0.062) | 29.4 (1.0) | 1.3 (0.4) | 8.4 (5.9) |
| D2-DPO (Borso et al., 2025) | 0.500 (0.051) | 66.4 (0.3) | 0.909 (0.005) | 93.6 (0.8) | 61.0 (0.5) | 41.7 (7.4) |
| VRPO (Zhu et al., 2025a) | 0.548 (0.032) | 61.1 (0.1) | 0.883 (0.004) | 93.5 (0.7) | 56.6 (0.3) | 39.1 (9.3) |
| DDPP-IS (Zhu et al., 2025a) | -0.130 (0.047) | 46.7 (0.8) | **0.829** (0.008) | 89.3 (0.6) | 43.3 (0.5) | 24.3 (7.6) |
| DRAKES (Wang et al., 2024) | 1.095 (0.026) | 86.4 (0.2) | 0.918 (0.006) | 91.8 (0.5) | **78.6** (0.7) | 33.3 (6.4) |
| diffu-GRPO (Zhao et al., 2025) | 1.286 (0.021) | 76.8 (0.3) | 1.192 (0.005) | 57.1 (0.8) | 37.2 (1.4) | 40.0 (7.8) |
| SDPO | **1.400** (0.014) | **87.1** (0.01) | 0.938 (0.005) | 88.9 (0.3) | 75.5 (0.3) | **42.3** (6.5) |

**Experiment setup.** The pre-trained diffusion model uses the ProteinMPNN (Dauparas et al., 2022) architecture and is trained using the methodology from (Campbell et al., 2024) on the PDB training dataset from Dauparas et al. (2022). The finetuning and evaluation reward models are trained on different splits of the Megascale (Tsuboyama et al., 2023) dataset. We take all checkpoints directly from Wang et al. (2024). For finetuning our model, we use the curated Megascale training dataset from Wang et al. (2024), which consists of ∼500k sequences with stability measurements.

**Metrics.** We use the following metrics (Wang et al., 2024) to evaluate the stability and naturalness of the generated protein sequences. **1.** *Pred-ddG.* The evaluation reward model predicts the ddG (change in Gibbs free energy) of a sequence, which is a measure of the sequence's stability. The finetuning dataset does not overlap with the evaluation dataset, so the model does not train on proteins used for evaluation. **2.** *scRMSD.* The self-consistency root mean square deviation (scRMSD) measures the ability of a sequence to fold into the desired structure. We use the pre-trained ESMFold (Lin et al., 2023) model to compute the RMSD between the sequence's predicted 3D structure and the original backbone structure. **3.** *Success rate.* We compute the success rate as the proportion of generated sequences with Pred-ddG $> 0$ and scRMSD $< 2$. **4.** *Entropy.* The sequence entropy is computed to measure the sample diversity.

**Results.** Our method is able to generate sequences with high stability that still remain in-distribution. We significantly outperform all baselines in the predicted ddG for stability, showing strong reward optimization ability, while still producing natural-like samples with scRMSD values and overall success rate comparable to the state-of-the-art *DRAKES* method. The policy-gradient based method diffu-GRPO exhibits significant reward over-optimization with severe drop in metrics like Success Rate. Notably, the inverse folding problem is particularly difficult due to lack of labeled data in the curated Megascale dataset (only several hundred distinct 3D structure backbones). During evaluation, the model conditions on new backbone configurations not seen during training. Thus, our method is still able to generate high reward samples without over-optimizing in a limited-data setting.

## 5.3 LANGUAGE MODELING

Crucially, we also apply our approach to a large-scale discrete diffusion for natural language modeling, demonstrating its efficacy towards preference optimization of large language diffusion models.

**Experiment setup.** We employ LLaDA-8B-Instruct (Nie et al., 2025), a large-scale instruction-tuned chat model based on the masked diffusion framework, as the reference model. We use UltraFeedback (Cui et al., 2023) dataset annotated by Meng et al. (2024) as the preference dataset, and finetune the model on 8 Nvidia A100 GPUs. Detailed hyperparameters are deferred to Appendix.

**Benchmarks and metrics.** We compare our finetuned model against the reference model on three important language model benchmarks. **1.** *GSM8K* (Cobbe et al., 2021), which benchmarks the math and reasoning capability of the model on graduate school math problems. The metric is the average accuracy of the answers. **2.** *IFEval* (Zhou et al., 2023), which measures the model's capabil-

Table 3: Results on finetuning LLaDA-8B-Instruct using dataset from Cui et al. (2023).

| | | Instruct | D2-DPO | diffu-GRPO | SDPO |
|---|---|---|---|---|---|
| Alpaca- | LC (%) | 10.6 | 12.1 | 12.6 | **14.2** |
| Eval 2.0 | WR (%) | 6.8 | 7.5 | 7.8 | **8.7** |
| | GSM8K | 78.6 | 78.1 | 80.5 | **81.2** |
| | IFEval | 52.9 | 53.8 | 53.5 | **55.1** |

ity of following human natural language instructions. We report IFEval score, the average of prompt and instruction-level strict-accuracy. **3.** *AlpacaEval 2.0* (Li et al., 2023; Dubois et al., 2024) that evaluates the chat response quality by comparing against certain baseline model on a suite of prompts.

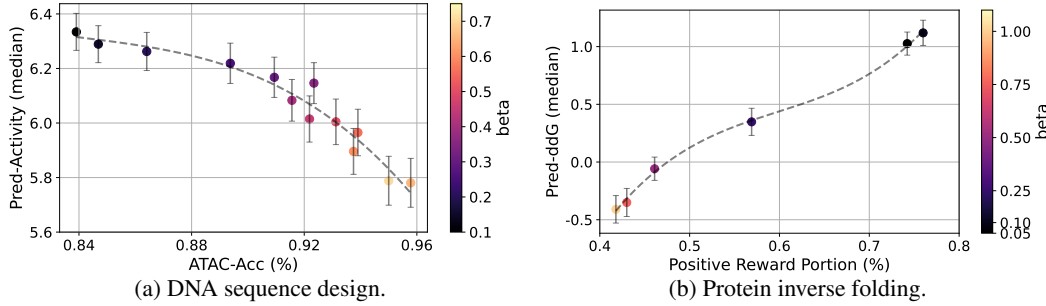

(a) DNA sequence design.  (b) Protein inverse folding.

Figure 2: Ablation studies of $\beta$ in (a) DNA design, and (b) protein inverse folding experiment.

The metrics on this benchmark are the winrate (WR) and length-controlled (LC) winrate against GPT-4-Preview-1106.

**Results.** The benchmark results are presented in Table 3. By finetuning LLaDA-8B-Instruct using our proposed SDPO, we observe a consistent and remarkable enhancement across all three benchmarks, which underscores the efficacy of SDPO towards promoting the capability of mathematical reasoning, instruction following, and chat quality of the discrete diffusion language model. Notably, our approach improves GSM8K score from 78.6 to 81.2, surpassing the score of LLaMA-3-8B post-trained with RL (c.f. Nie et al. (2025)). Furthermore, we obtain a relative improvement of 30.9% averaged across LC and WR on AlpacaEval 2.0 benchmark, demonstrating the applicability of SDPO for building helpful discrete diffusion-based chatbot. Our results on language modeling tasks open up new possibility towards building performant large language diffusion models through preference optimization.

## 5.4 ABLATION STUDY

**The effect of $\beta$.** We study the effect of $\beta$ in aligning models. As shown in Eq. 5, choosing a smaller $\beta$ leads to weaker regularization and tunes the model further away from the pretrained reference distribution. We verify this by two ablation studies on DNA sequence design and protein inverse folding, fixing all hyperparameters except $\beta$. Fig. 2 shows that a lower $\beta$ value results in stronger reward guidance, resulting in greater *Pred-Activity* for DNA design, and greater *Pred-ddG* values for protein inverse folding. Conversely, a larger $\beta$ poses more regularization to the model and thus the reward remains closer to the pretrained reference model. However, choosing too small a $\beta$ may also steer the model too far away from the reference model and result in unnatural sequences. As shown in Fig. 2(a), the *ATAC-Acc* of the generated DNA sequences decreases as we over-optimizes *Pred-Activity* with a small $\beta$, despite their being positively correlated for natural DNA sequences.

**The effect of $N$.** We first investigate the effect of the sample size $N$. The results in protein inverse folding task without any iterative labeling are presented in Table 4. Notably, we observe that as the value of $N$ gradually increases, we effectively reduces the variance in Monte-Carlo estimate performed by Eq. 10, which is further supported by the increasing trend in *Pred-ddG*. In particular, compared with $N = 2$ which reflects the pairwise preference data setting adopted in DPO,

Table 4: Effect of sample size $N$ in the inverse protein folding task.

| $N$ | *Pred-ddG* | *Positive Reward Prop.* |
|-----|-----------|------------------------|
| 2   | 0.529     | 0.624                  |
| 10  | 0.924     | 0.749                  |
| 25  | 1.119     | 0.759                  |
| 100 | 1.061     | 0.765                  |

leveraging a comparatively larger $N$ is more beneficial. The performance plateaus as $N$ further increases from 25 to 100, which is empirically not as favorable due to the memory overhead incurred.

**Iterative labeling.** In § 4.4 we additionally introduce a variant of our SDPO that leverages iterative labeling to enhance performance. Specifically, during training we iteratively generate 10,000 samples from the model and label them using the reward model in the DNA experiment. We then optimize the model on these labeled samples using the same objective. We demonstrate the advantage of such an approach in Fig. 3. Compared with the baseline that does not scale up the labeling on latest samples but always on samples from the original model, we observe consistent increment over 2 rounds of iterative labeling. In particular, the predicted DNA

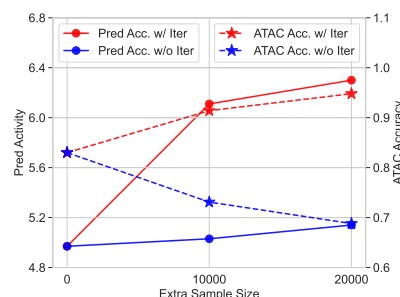

Figure 3: Iterative labeling.

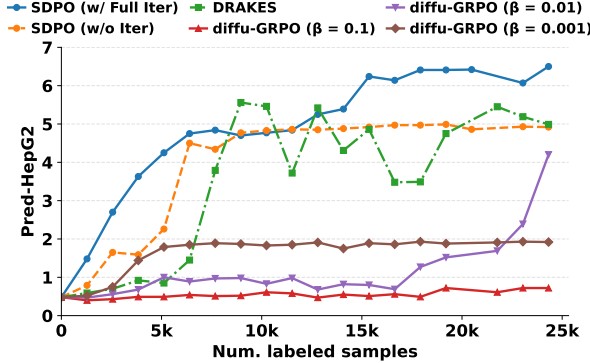
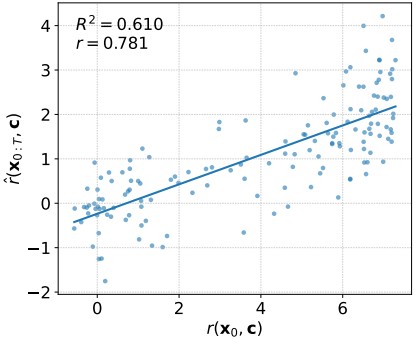

(a) Reward convergence *w.r.t.* the number of labeled samples.

(b) Correlation analysis between $r(\mathbf{x}_0, \mathbf{c})$ and the induced trajectory reward $\hat{r}(\mathbf{x}_{0:T}, \mathbf{c})$.

Figure 4: (a) The reward curve *w.r.t.* the number of labeled samples throughout training. (b) The correlation analysis between the induced trajectory reward $\hat{r}(\mathbf{x}_{0:T}, \mathbf{c})$ and the clean reward $r(\mathbf{x}_0, \mathbf{c})$.

activity improves by a significant margin for SDPO with iterative labeling while the counterpart struggles in predicted activity while also encountering a drop in ATAC accuracy, possibly due to overfitting. Furthermore, our approach is also remarkably more labeling efficient compared with DRAKES that uses 128,000 additional labeling on the DNA task. The result implies that, on certain tasks when the reward model is available, performing SDPO in an iterative manner with reward labeling will lead to improved performance.

**Convergence rate comparison.** We also perform a systematic head-to-head comparison of the reward convergence speed in Fig. 4a, where we plot the reward curve *w.r.t.* the number of labeled samples throughout the training process. For a fair comparison with the on-policy baselines DRAKES and diffu-GRPO, we implement a variant of SDPO, *i.e.*, SDPO w/ Full Iter, that performs iterative labeling after *each* training step. Notably, our SDPO with full iterative labeling achieves 6.2 Pred-HepG2 using only 15k labeled samples, while DRAKES and diffu-GRPO only achieve 5.6 and 4.2 Pred-HepG2 with 25k labeled samples. SDPO without iterative labeling also exhibits fast convergence and high reward efficiency.

**Reward correlation analysis.** Here we provide more in-depth analysis on the DNA task regarding the relationship between the reward $\hat{r}(\mathbf{x}_{0:T}, \mathbf{c})$ defined in Theorem 4.1 and the original reward $r(\mathbf{x}_0, \mathbf{c})$. Specifically, we sample 50 trajectories from the pretrained model, the model after first stage training, and the final model respectively, leading to 150 trajectories in total $\{\mathbf{x}_{0:T}^{(i)}\}_{i=1}^{150}$. For each trajectory $\mathbf{x}_{0:T}^{(i)}$, we evaluate its chain reward $\hat{r}^{(i)} = \hat{r}(\mathbf{x}_{0:T}^{(i)}, \mathbf{c})$ using an unbiased MC estimator (see Appendix B.5) and the original reward $r^{(i)} = r(\mathbf{x}_0^{(i)}, \mathbf{c})$. We then perform linear correlation analysis for the set of datapoints $\{(r^{(i)}, \hat{r}^{(i)})\}_{i=1}^{150}$ and show the plot in Fig. 4b. Despite the MC estimation, we observe a relatively strong positive correlation between the rewards with a Pearson correlation of 0.781. This indicates that the chain reward can be approximately viewed as $\hat{r} \approx a \cdot r + b$, which draws an interesting connection between the reward-tilted distributions: $p_{\text{ref}}(\mathbf{x}_{0:T}|\mathbf{c}) \exp(\frac{1}{\beta}\hat{r}(\mathbf{x}_{0:T}, \mathbf{c})) \approx p_{\text{ref}}(\mathbf{x}_{0:T}|\mathbf{c}) \exp(\frac{a}{\beta} \cdot r(\mathbf{x}_0, \mathbf{c}) + \frac{b}{\beta}) \propto p_{\text{ref}}(\mathbf{x}_{0:T}|\mathbf{c}) \exp(\frac{1}{\beta/a} \cdot r(\mathbf{x}_0, \mathbf{c}))$. Therefore, this empirical investigation interestingly reveals that our trajectory-level chain reward is an effective surrogate of the original reward on $\mathbf{x}_0$ and, more importantly, our optimal reward-tilted distribution is also a decent approximation of the original optimal reward-tilted distribution without notable bias introduced.

# 6 CONCLUSION

We present SDPO for preference optimization of discrete diffusion models by decomposing diffusion trajectory alignment into a set of subproblems for each diffusion step. Crucially, we propose to align the posterior $\hat{p}_\theta(\mathbf{x}_0|\mathbf{x}_t)$ for each step and draw an equivalence between the two objectives, with which we further derive a principled loss function. Experiments on a wide range of tasks including DNA sequence design, protein inverse folding, and language modeling consistently verify the efficacy of SDPO, showing its potential towards building performant and applicable discrete diffusion models.

## ETHICS STATEMENT

All authors have read and are committed to comply with the ICLR Code of Ethics (https://iclr.cc/public/CodeOfEthics). We present a principled approach for discrete diffusion model alignment via stepwise decomposition. The goal is to develop a fundamental algorithm for alignment of discrete diffusion models, where we do not find major ethical concerns that need to highlight.

## REPRODUCIBILITY STATEMENT

We have presented the detailed experimental setup in Sec. 5 and Appendix B. We also include the code in the supplementary material to ensure reproducibility.

## ACKNOWLEDGMENT

We thank the anonymous reviewers for their feedback on improving the manuscript. This work was supported by ARO (W911NF-21-1-0125), ONR (N00014-23-1-2159), and the CZ Biohub.

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

## A PROOFS

### A.1 PROOF OF THEOREM 4.1

**Theorem 4.1.** *The joint $p^*(\mathbf{x}_{0:T}|\mathbf{c})$ induced by the optimal solutions $\{\hat{p}^*(\mathbf{x}_0|\mathbf{x}_t, \mathbf{c})\}_{t=1}^T$ of Eq. 6 is also the optimal solution of the trajectory alignment objective in Eq. 5, with the chain reward*
$$\hat{r}(\mathbf{x}_{0:T}, \mathbf{c}) = \beta \sum_{t=1}^T r_t(\mathbf{x}_{t-1}, \mathbf{x}_t, \mathbf{c}) \text{ where } r_t(\mathbf{x}_{t-1}, \mathbf{x}_t, \mathbf{c}) = \log \frac{\mathbb{E}_{p'_{\mathrm{ref}}(\mathbf{x}_0|\mathbf{x}_{t-1}, \mathbf{x}_t, \mathbf{c})}\left[\exp\left(\frac{1}{\beta_t} r(\mathbf{x}_0, \mathbf{c})\right)\right]}{\mathbb{E}_{\hat{p}_{\mathrm{ref}}(\mathbf{x}_0|\mathbf{x}_t, \mathbf{c})}\left[\exp\left(\frac{1}{\beta_t} r(\mathbf{x}_0, \mathbf{c})\right)\right]}.$$

*Proof.* Leveraging Eq. 4, the optimal solution for each subproblem in Eq. 6 is given by

$$\hat{p}^*(\mathbf{x}_0|\mathbf{x}_t, \mathbf{c}) = \frac{1}{Z_t(\mathbf{x}_t, \mathbf{c})} \hat{p}_{\mathrm{ref}}(\mathbf{x}_0|\mathbf{x}_t, \mathbf{c}) \exp\left(\frac{1}{\beta_t} r(\mathbf{x}_0, \mathbf{c})\right), \quad \forall 1 \le t \le T, \tag{12}$$

where $Z_t(\mathbf{x}_t, \mathbf{c}) = \sum_{\mathbf{x}_0} \hat{p}_{\mathrm{ref}}(\mathbf{x}_0|\mathbf{x}_t, \mathbf{c}) \exp\left(\frac{1}{\beta_t} r(\mathbf{x}_0, \mathbf{c})\right) = \mathbb{E}_{\hat{p}_{\mathrm{ref}}(\mathbf{x}_0|\mathbf{x}_t, \mathbf{c})}\left[\exp\left(\frac{1}{\beta_t} r(\mathbf{x}_0, \mathbf{c})\right)\right]$. The transition kernels $p^*(\mathbf{x}_{t-1}|\mathbf{x}_t, \mathbf{c})$ induced by the solutions can be derived as

$$p^*(\mathbf{x}_{t-1}|\mathbf{x}_t, \mathbf{c}) = \sum_{\mathbf{x}_0} \hat{p}^*(\mathbf{x}_0|\mathbf{x}_t, \mathbf{c}) q(\mathbf{x}_{t-1}|\mathbf{x}_0, \mathbf{x}_t), \tag{13}$$

$$= \sum_{\mathbf{x}_0} \frac{1}{Z_t(\mathbf{x}_t, \mathbf{c})} \hat{p}_{\mathrm{ref}}(\mathbf{x}_0|\mathbf{x}_t, \mathbf{c}) \exp\left(\frac{1}{\beta_t} r(\mathbf{x}_0, \mathbf{c})\right) q(\mathbf{x}_{t-1}|\mathbf{x}_0, \mathbf{x}_t), \tag{14}$$

$$= p_{\mathrm{ref}}(\mathbf{x}_{t-1}|\mathbf{x}_t, \mathbf{c}) \sum_{\mathbf{x}_0} \frac{\hat{p}_{\mathrm{ref}}(\mathbf{x}_0|\mathbf{x}_t, \mathbf{c}) q(\mathbf{x}_{t-1}|\mathbf{x}_0, \mathbf{x}_t)}{Z_t(\mathbf{x}_t, \mathbf{c}) p_{\mathrm{ref}}(\mathbf{x}_{t-1}|\mathbf{x}_t)} \exp\left(\frac{1}{\beta_t} r(\mathbf{x}_0, \mathbf{c})\right), \tag{15}$$

$$= \frac{1}{Z_t(\mathbf{x}_t, \mathbf{c})} p_{\mathrm{ref}}(\mathbf{x}_{t-1}|\mathbf{x}_t, \mathbf{c}) \sum_{\mathbf{x}_0} \frac{\hat{p}_{\mathrm{ref}}(\mathbf{x}_0|\mathbf{x}_t, \mathbf{c}) q(\mathbf{x}_{t-1}|\mathbf{x}_0, \mathbf{x}_t)}{p_{\mathrm{ref}}(\mathbf{x}_{t-1}|\mathbf{x}_t, \mathbf{c})} \exp\left(\frac{1}{\beta_t} r(\mathbf{x}_0, \mathbf{c})\right), \tag{16}$$

$$= \frac{1}{Z_t(\mathbf{x}_t, \mathbf{c})} p_{\mathrm{ref}}(\mathbf{x}_{t-1}|\mathbf{x}_t, \mathbf{c}) \mathbb{E}_{p'_{\mathrm{ref}}(\mathbf{x}_0|\mathbf{x}_{t-1}, \mathbf{x}_t, \mathbf{c})}\left[\exp\left(\frac{1}{\beta_t} r(\mathbf{x}_0, \mathbf{c})\right)\right], \tag{17}$$

where $p'_{\mathrm{ref}}(\mathbf{x}_0|\mathbf{x}_{t-1}, \mathbf{x}_t, \mathbf{c}) \coloneqq \frac{\hat{p}_{\mathrm{ref}}(\mathbf{x}_0|\mathbf{x}_t, \mathbf{c}) q(\mathbf{x}_{t-1}|\mathbf{x}_0, \mathbf{x}_t)}{p_{\mathrm{ref}}(\mathbf{x}_{t-1}|\mathbf{x}_t, \mathbf{c})}$. Specifically, Eq. 13 holds due to the $\mathbf{x}_0$-parameterization of the transition kernel (see Austin et al. (2021); Shi et al. (2024)). Notably it is straightforward to verify that $p'_{\mathrm{ref}}(\mathbf{x}_0|\mathbf{x}_{t-1}, \mathbf{x}_t, \mathbf{c})$ is a properly normalized distribution since $\sum_{\mathbf{x}_0} p'_{\mathrm{ref}}(\mathbf{x}_0|\mathbf{x}_{t-1}, \mathbf{x}_t, \mathbf{c}) = \frac{\sum_{\mathbf{x}_0} \hat{p}_{\mathrm{ref}}(\mathbf{x}_0|\mathbf{x}_t, \mathbf{c}) q(\mathbf{x}_{t-1}|\mathbf{x}_0, \mathbf{x}_t)}{p_{\mathrm{ref}}(\mathbf{x}_{t-1}|\mathbf{x}_t, \mathbf{c})} = \frac{p(\mathbf{x}_{t-1}|\mathbf{x}_t, \mathbf{c})}{p(\mathbf{x}_{t-1}|\mathbf{x}_t, \mathbf{c})} = 1$.

Plugging it back into the Markovian factorization of the reverse process, we arrive at

$$p^*(\mathbf{x}_{0:T}|\mathbf{c}) = p(\mathbf{x}_T) \prod_{t=1}^{t=T} p^*(\mathbf{x}_{t-1}|\mathbf{x}_t, \mathbf{c}), \tag{18}$$

$$= p(\mathbf{x}_T) \prod_{t=1}^{t=T} \left( \frac{1}{Z_t(\mathbf{x}_t, \mathbf{c})} p_{\text{ref}}(\mathbf{x}_{t-1}|\mathbf{x}_t, \mathbf{c}) \mathbb{E}_{p'_{\text{ref}}(\mathbf{x}_0|\mathbf{x}_{t-1}, \mathbf{x}_t, \mathbf{c})} [r(\mathbf{x}_0, \mathbf{c})] \right), \tag{19}$$

$$= p(\mathbf{x}_T) \prod_{t=1}^{t=T} p_{\text{ref}}(\mathbf{x}_{t-1}|\mathbf{x}_t, \mathbf{c}) \prod_{t=1}^{t=T} \frac{\mathbb{E}_{p'_{\text{ref}}(\mathbf{x}_0|\mathbf{x}_{t-1}, \mathbf{x}_t, \mathbf{c})} \left[ \exp\left( \frac{1}{\beta_t} r(\mathbf{x}_0, \mathbf{c}) \right) \right]}{Z_t(\mathbf{x}_t, \mathbf{c})}, \tag{20}$$

$$= p_{\text{ref}}(\mathbf{x}_{0:T}|\mathbf{c}) \exp\left( \sum_{t=1}^{T} \log \frac{\mathbb{E}_{p'_{\text{ref}}(\mathbf{x}_0|\mathbf{x}_{t-1}, \mathbf{x}_t, \mathbf{c})} \left[ \exp\left( \frac{1}{\beta_t} r(\mathbf{x}_0, \mathbf{c}) \right) \right]}{Z_t(\mathbf{x}_t, \mathbf{c})} \right), \tag{21}$$

$$= p_{\text{ref}}(\mathbf{x}_{0:T}|\mathbf{c}) \exp\left( \sum_{t=1}^{T} \log \frac{\mathbb{E}_{p'_{\text{ref}}(\mathbf{x}_0|\mathbf{x}_{t-1}, \mathbf{x}_t, \mathbf{c})} \left[ \exp\left( \frac{1}{\beta_t} r(\mathbf{x}_0, \mathbf{c}) \right) \right]}{\mathbb{E}_{p_{\text{ref}}(\mathbf{x}_0|\mathbf{x}_t, \mathbf{c})} \left[ \exp\left( \frac{1}{\beta_t} r(\mathbf{x}_0, \mathbf{c}) \right) \right]} \right), \tag{22}$$

$$= p_{\text{ref}}(\mathbf{x}_{0:T}|\mathbf{c}) \exp\left( \sum_{t=1}^{T} r_t(\mathbf{x}_{t-1}, \mathbf{x}_t, \mathbf{c}) \right), \tag{23}$$

$$= p_{\text{ref}}(\mathbf{x}_{0:T}|\mathbf{c}) \exp\left( \frac{1}{\beta} \cdot \underbrace{\beta \sum_{t=1}^{T} r_t(\mathbf{x}_{t-1}, \mathbf{x}_t, \mathbf{c})}_{\hat{r}(\mathbf{x}_{0:T}, \mathbf{c})} \right), \tag{24}$$

where $r_t(\mathbf{x}_{t-1}, \mathbf{x}_t, \mathbf{c}) = \log \frac{\mathbb{E}_{p'_{\text{ref}}(\mathbf{x}_0|\mathbf{x}_{t-1}, \mathbf{x}_t, \mathbf{c})} \left[ \exp\left( \frac{1}{\beta_t} r(\mathbf{x}_0, \mathbf{c}) \right) \right]}{\mathbb{E}_{\hat{p}_{\text{ref}}(\mathbf{x}_0|\mathbf{x}_t, \mathbf{c})} \left[ \exp\left( \frac{1}{\beta_t} r(\mathbf{x}_0, \mathbf{c}) \right) \right]}$. Eq. 23 directly implies that the induced distribution $p^*(\mathbf{x}_{0:T}|\mathbf{c})$ is the optimal solution of the trajectory alignment objective in Eq. 5 with $\hat{r}(\mathbf{x}_{0:T}, \mathbf{c}) = \beta \sum_{t=1}^{T} r_t(\mathbf{x}_{t-1}, \mathbf{x}_t, \mathbf{c})$, which concludes the proof. □

### A.2 PROOF OF PROPOSITION 4.2

**Proposition 4.2.** *Let $\theta^* = \arg\min \mathcal{L}_t(\theta)$ defined in Eq. 7. Then $\hat{p}_{\theta^*}(\mathbf{x}_0|\mathbf{x}_t, \mathbf{c})$ is the optimal solution of the stepwise alignment objective in Eq. 6.*

*Proof.* Recall the definition of $\mathcal{L}_t(\theta)$ in Eq. 7:

$$\mathcal{L}_t(\theta) := \mathbb{E}_{\mathbf{x}_t, \mathbf{c}} \left[ D_{\text{KL}} \left[ \tilde{p}_r(\mathbf{x}_0|\mathbf{x}_t, \mathbf{c}) \| \tilde{p}_\theta(\mathbf{x}_0|\mathbf{x}_t, \mathbf{c}) \right] \right]. \tag{25}$$

Since the KL-divergence is minimized when the two distributions are exactly matched, we have that the optimal $\theta^*$ satisfies

$$\tilde{p}_r(\mathbf{x}_0|\mathbf{x}_t, \mathbf{c}) = \tilde{p}_{\theta^*}(\mathbf{x}_0|\mathbf{x}_t, \mathbf{c}). \tag{26}$$

By leveraging the definition of $\tilde{p}_r$ and $\tilde{p}_\theta$, we have

$$\frac{1}{Z_r(\mathbf{x}_t, \mathbf{c})} \hat{p}_{\text{ref}}(\mathbf{x}_0|\mathbf{x}_t, \mathbf{c}) \exp(r(\mathbf{x}_0, \mathbf{c})) = \frac{1}{Z_{\theta^*}(\mathbf{x}_t, \mathbf{c})} \hat{p}_{\text{ref}}(\mathbf{x}_0|\mathbf{x}_t, \mathbf{c})^{(1-\beta_t)} \hat{p}_{\theta^*}(\mathbf{x}_0|\mathbf{x}_t, \mathbf{c})^{\beta_t}, \tag{27}$$

which simplifies to

$$\left( \frac{\hat{p}_{\theta^*}(\mathbf{x}_0|\mathbf{x}_t, \mathbf{c})}{\hat{p}_{\text{ref}}(\mathbf{x}_0|\mathbf{x}_t, \mathbf{c})} \right)^{\beta_t} = \frac{Z_{\theta^*}(\mathbf{x}_t, \mathbf{c})}{Z_r(\mathbf{x}_t, \mathbf{c})} \exp(r(\mathbf{x}_0, \mathbf{c})), \tag{28}$$

□

and finally gives us

$$\hat{p}_{\theta^*}(\mathbf{x}_0|\mathbf{x}_t, \mathbf{c}) = \frac{1}{Z'(\mathbf{x}_t, \mathbf{c})} \hat{p}_{\text{ref}}(\mathbf{x}_0|\mathbf{x}_t, \mathbf{c}) \exp(\frac{1}{\beta_t} r(\mathbf{x}_0, \mathbf{c})), \tag{29}$$

where $Z'(\mathbf{x}_t, \mathbf{c}) = \left( \frac{Z_r(\mathbf{x}_t, \mathbf{c})}{Z_{\theta^*}(\mathbf{x}_t, \mathbf{c})} \right)^{1/\beta_t}$. The proof is therefore completed.

### A.3 PROOF OF THE LOSS IN EQ. 7

Here we provide the detailed derivation on how to derive Eq. 7 from $\mathcal{L}_t(\theta)$ step-by-step.

We start from the definition of $\mathcal{L}_t(\theta)$ in Eq. 6:

$$\mathcal{L}_t(\theta) = \mathbb{E}_{\mathbf{x}_t,\mathbf{c}} \left[ D_{\mathrm{KL}} \left[ \tilde{p}_r(\mathbf{x}_0|\mathbf{x}_t,\mathbf{c}) \| \tilde{p}_\theta(\mathbf{x}_0|\mathbf{x}_t,\mathbf{c}) \right] \right], \tag{30}$$

$$= \mathbb{E}_{\mathbf{x}_t,\mathbf{c},\mathbf{x}_0 \sim \tilde{p}_r(\mathbf{x}_0|\mathbf{x}_t,\mathbf{c})} \left[ \log \left( \frac{\tilde{p}_r(\mathbf{x}_0|\mathbf{x}_t,\mathbf{c})}{\tilde{p}_\theta(\mathbf{x}_0|\mathbf{x}_t,\mathbf{c})} \right) \right], \tag{31}$$

$$= \mathbb{E}_{\mathbf{c},(\mathbf{x}_0,\mathbf{x}_t) \sim \tilde{p}_r(\mathbf{x}_0,\mathbf{x}_t|\mathbf{c})} \left[ \log \left( \frac{\tilde{p}_r(\mathbf{x}_0|\mathbf{x}_t,\mathbf{c})}{\tilde{p}_\theta(\mathbf{x}_0|\mathbf{x}_t,\mathbf{c})} \right) \right], \tag{32}$$

$$= \mathbb{E}_{\mathbf{c},(\mathbf{x}_0,\mathbf{x}_t) \sim \hat{p}_{\mathrm{ref}}(\mathbf{x}_0,\mathbf{x}_t|\mathbf{c})} \left[ \frac{\tilde{p}_r(\mathbf{x}_0,\mathbf{x}_t|\mathbf{c})}{\hat{p}_{\mathrm{ref}}(\mathbf{x}_0,\mathbf{x}_t|\mathbf{c})} \log \left( \frac{\tilde{p}_r(\mathbf{x}_0|\mathbf{x}_t,\mathbf{c})/\hat{p}_{\mathrm{ref}}(\mathbf{x}_0|\mathbf{x}_t,\mathbf{c})}{\tilde{p}_\theta(\mathbf{x}_0|\mathbf{x}_t,\mathbf{c})/\hat{p}_{\mathrm{ref}}(\mathbf{x}_0|\mathbf{x}_t,\mathbf{c})} \right) \right], \tag{33}$$

$$= \mathbb{E}_{\mathbf{c},(\mathbf{x}_0,\mathbf{x}_t) \sim \hat{p}_{\mathrm{ref}}(\mathbf{x}_0,\mathbf{x}_t|\mathbf{c})} \left[ \frac{\exp(r(\mathbf{x}_0,\mathbf{c}))}{\mathbb{E}_{\hat{p}_{\mathrm{ref}}(\mathbf{x}_0,\mathbf{x}_t|\mathbf{c})} \exp(r(\mathbf{x}_0,\mathbf{c}))} \log \left( \frac{\tilde{p}_r(\mathbf{x}_0|\mathbf{x}_t,\mathbf{c})/\hat{p}_{\mathrm{ref}}(\mathbf{x}_0|\mathbf{x}_t,\mathbf{c})}{\tilde{p}_\theta(\mathbf{x}_0|\mathbf{x}_t,\mathbf{c})/\hat{p}_{\mathrm{ref}}(\mathbf{x}_0|\mathbf{x}_t,\mathbf{c})} \right) \right], \tag{34}$$

$$= \mathbb{E}_{\mathbf{c},(\mathbf{x}_0,\mathbf{x}_t) \sim \hat{p}_{\mathrm{ref}}(\mathbf{x}_0,\mathbf{x}_t|\mathbf{c})} \left[ \frac{\exp(r(\mathbf{x}_0,\mathbf{c}))}{\mathbb{E}_{\hat{p}_{\mathrm{ref}}(\mathbf{x}_0|\mathbf{c})} \exp(r(\mathbf{x}_0,\mathbf{c}))} \log \left( \frac{\tilde{p}_r(\mathbf{x}_0|\mathbf{x}_t,\mathbf{c})/\hat{p}_{\mathrm{ref}}(\mathbf{x}_0|\mathbf{x}_t,\mathbf{c})}{\tilde{p}_\theta(\mathbf{x}_0|\mathbf{x}_t,\mathbf{c})/\hat{p}_{\mathrm{ref}}(\mathbf{x}_0|\mathbf{x}_t,\mathbf{c})} \right) \right], \tag{35}$$

$$= -\mathbb{E}_{\mathbf{c},(\mathbf{x}_0,\mathbf{x}_t) \sim \hat{p}_{\mathrm{ref}}(\mathbf{x}_0,\mathbf{x}_t|\mathbf{c})} \left[ \frac{\exp(r(\mathbf{x}_0,\mathbf{c}))}{Z_r(\mathbf{c})} \log \left( \frac{1}{Z_\theta^t(\mathbf{x}_t,\mathbf{c},\beta_t)} \left( \frac{p_\theta(\mathbf{x}_0|\mathbf{x}_t,\mathbf{c})}{\hat{p}_{\mathrm{ref}}(\mathbf{x}_0|\mathbf{x}_t,\mathbf{c})} \right)^{\beta_t} \right) \right] + C, \tag{36}$$

where the last step extracts the constant $C$ out of the numerator of $\log$ since it is irrelevant to $\theta$.

Recalling the definition of the implicit reward, which is given by $\tilde{r}_\theta(\mathbf{x}_0,\mathbf{x}_t,\mathbf{c},\beta_t) = \beta_t(\log p_\theta(\mathbf{x}_0|\mathbf{x}_t,\mathbf{c}) - \log \hat{p}_{\mathrm{ref}}(\mathbf{x}_0|\mathbf{x}_t,\mathbf{c}))$, we have that $\left( \frac{\hat{p}_\theta(\mathbf{x}_0|\mathbf{x}_t,\mathbf{c})}{\hat{p}_{\mathrm{ref}}(\mathbf{x}_0|\mathbf{x}_t,\mathbf{c})} \right)^{\beta_t} = \exp(\tilde{r}_\theta(\mathbf{x}_0,\mathbf{x}_t,\mathbf{c},\beta_t))$. Therefore, we can further simplify

$$\mathcal{L}_t(\theta) = -\mathbb{E}_{\mathbf{c},(\mathbf{x}_0,\mathbf{x}_t) \sim \hat{p}_{\mathrm{ref}}(\mathbf{x}_0,\mathbf{x}_t|\mathbf{c})} \left[ \frac{\exp(r(\mathbf{x}_0,\mathbf{c}))}{Z_r(\mathbf{c})} \log \left( \frac{\exp(\tilde{r}_\theta(\mathbf{x}_0,\mathbf{x}_t,\mathbf{c},\beta_t))}{Z_\theta^t(\mathbf{x}_t,\mathbf{c},\beta_t)} \right) \right] + C, \tag{37}$$

where $C$ is a constant irrelevant to $\theta$ and $Z_r(\mathbf{c})$ and $Z_\theta^t(\mathbf{x}_t,\mathbf{c},\beta_t)$ are the partition functions.

### A.4 DERIVATION OF SDPO LOSS IN THE DPO SETTING

Recall our proposed loss function $\mathcal{L}(\theta)$:

$$\mathcal{L}(\theta) = -\mathbb{E}_{t,\mathbf{c},\mathbf{x}_0,q(\mathbf{x}_t|\mathbf{x}_0)} \sum_{i=1}^{N} \left( \frac{\exp(r(\mathbf{x}_0^{(i)},\mathbf{c}))}{\sum_{j=1}^{N} \exp(r(\mathbf{x}_0^{(j)},\mathbf{c}))} \cdot \log \frac{\exp(\tilde{r}_\theta(\mathbf{x}_0^{(i)},\mathbf{x}_t^{(i)},\mathbf{c},\beta_t))}{\sum_{j=1}^{N} \exp(\tilde{r}_\theta(\mathbf{x}_0^{(j)},\mathbf{x}_t^{(j)},\mathbf{c},\beta_t))} \right), \tag{38}$$

with

$$\tilde{r}_\theta(\mathbf{x}_0,\mathbf{x}_t,\mathbf{c},\beta_t) = \beta \left( \frac{\log(\mathbf{x}_0^\top \mathbf{f}_\theta(\mathbf{x}_t,t,\mathbf{c}))}{w(t)} - \frac{\log(\mathbf{x}_0^\top \mathbf{f}_{\mathrm{ref}}(\mathbf{x}_t,t,\mathbf{c}))}{w(t)} \right). \tag{39}$$

Here we derive a specific instance of $\mathcal{L}(\theta)$ in the DPO pairwise preference setting, and draw connection of it to Wallace et al. (2024).

In particular, in DPO preference pair setting for each context $\mathbf{c}$ there are two completions, namely, $N = 2$ in our case. Furthermore, one completion is labeled as the preferred (chosen) response $\mathbf{x}_0^{(w)}$ and the other as rejected sample $\mathbf{x}_0^{(l)}$. Since no explicit real-valued reward on the chosen and rejected sample is provided, the Bradley-Terry (BT) model (Bradley & Terry, 1952) is adopted, which corresponds to, in our case, setting $r(\mathbf{x}_0^{(w)},\mathbf{c}) = 0$ and $r(\mathbf{x}_0^{(l)},\mathbf{c}) = -\infty$. Under this specification,

$\mathcal{L}(\theta)$ is simplified as

$$\mathcal{L}(\theta) = -\mathbb{E}_{t,\mathbf{c},\mathbf{x}_0,q(\mathbf{x}_t|\mathbf{x}_0)}\Big[$$
$$\frac{\exp(r(\mathbf{x}_0^{(w)},\mathbf{c}))}{\exp(r(\mathbf{x}_0^{(w)},\mathbf{c})) + \exp(r(\mathbf{x}_0^{(l)},\mathbf{c}))} \cdot \log \frac{\exp(\tilde{r}_\theta(\mathbf{x}_0^{(w)},\mathbf{x}_t^{(w)}))}{\exp(\tilde{r}_\theta(\mathbf{x}_0^{(w)},\mathbf{x}_t^{(w)})) + \exp(\tilde{r}_\theta(\mathbf{x}_0^{(l)},\mathbf{x}_t^{(l)}))}$$
$$+ \frac{\exp(r(\mathbf{x}_0^{(l)},\mathbf{c}))}{\exp(r(\mathbf{x}_0^{(w)},\mathbf{c})) + \exp(r(\mathbf{x}_0^{(l)},\mathbf{c}))} \cdot \log \frac{\exp(\tilde{r}_\theta(\mathbf{x}_0^{(l)},\mathbf{x}_t^{(l)}))}{\exp(\tilde{r}_\theta(\mathbf{x}_0^{(w)},\mathbf{x}_t^{(w)})) + \exp(\tilde{r}_\theta(\mathbf{x}_0^{(l)},\mathbf{x}_t^{(l)}))}\Big],$$

$$= -\mathbb{E}_{t,\mathbf{c},\mathbf{x}_0,q(\mathbf{x}_t|\mathbf{x}_0)}\left[ \frac{1}{1+0} \log \frac{1}{1 + \exp(\tilde{r}_\theta(\mathbf{x}_0^{(l)},\mathbf{x}_t^{(l)}) - \tilde{r}_\theta(\mathbf{x}_0^{(w)},\mathbf{x}_t^{(w)}))} + 0 \right], \qquad (40)$$

$$= -\mathbb{E}_{t,\mathbf{c},\mathbf{x}_0,q(\mathbf{x}_t|\mathbf{x}_0)}\left[ \log \frac{1}{1 + \exp(\tilde{r}_\theta(\mathbf{x}_0^{(l)},\mathbf{x}_t^{(l)}) - \tilde{r}_\theta(\mathbf{x}_0^{(w)},\mathbf{x}_t^{(w)}))} \right], \qquad (41)$$

$$= -\mathbb{E}_{t,\mathbf{c},\mathbf{x}_0,q(\mathbf{x}_t|\mathbf{x}_0)} \log \sigma\left( \tilde{r}_\theta(\mathbf{x}_0^{(w)},\mathbf{x}_t^{(w)}) - \tilde{r}_\theta(\mathbf{x}_0^{(l)},\mathbf{x}_t^{(l)}) \right), \qquad (42)$$

where $\tilde{r}_\theta(\mathbf{x}_0^{(w)},\mathbf{x}_t^{(w)})$ is shorthand for $\tilde{r}_\theta(\mathbf{x}_0^{(w)},\mathbf{x}_t^{(w)},\mathbf{c},\beta_t)$ and similarly for the losing sample. Eq. 42 underscores an interesting connection of our loss to that of Wallace et al. (2024) specifically in the DPO setting, since both share the same form of negative logsigmoid over the margin between the implicit rewards of the winning and losing sample, with the difference being the definition of the implicit reward (Eq. 11), depending on whether using Gaussian diffusion (Wallace et al., 2024) or discrete diffusion as in this work. Notably, since we leverage a general formulation of stepwise decomposition that reduces the problem to a stepwise distribution matching objective, we are able to generalize to the setting with arbitrary number of samples and reward model, which is not revealed in Wallace et al. (2024).

## B   MORE EXPERIMENT DETAILS

### B.1   DNA SEQUENCE DESIGN

We use the pre-trained model and fine-tuning reward oracle from Wang et al. (2024) for finetuning with SDPO. In the first stage of finetuning, we train on the original enhancer dataset (also used for pre-training), without using re-labeled samples. In the next two stages, we generate 10000 samples from the finetuned model, label the samples with the reward model, and continue finetuning on the relabeled data. In all three stages, we use the original pre-trained checkpoint as the reference model. We provide hyperparameter configurations in Table 5.

Table 5: Detailed hyperparameters for DNA design task.

| Stage | # relabeled samples | $\beta$ | $N$ | # Epochs | Learning Rate |
|---|---|---|---|---|---|
| Stage 1 | 0 | 0.92 | 25 | 10 | 9e-5 |
| Stage 2 | 10k | 0.4 | 200 | 2 | 1e-5 |
| Stage 3 | 10k | 0.064 | 918 | 2 | 7.4e-6 |

### B.2   DETAILED HYPERPARAMETERS FOR PROTEIN INVERSE FOLDING TASK.

We also use the pre-trained model and fine-tuning reward oracle from Wang et al. (2024) for finetuning with SDPO. Likewise, we finetune on the original pre-training dataset without re-labeling any new samples. In the second stage, we re-label 12800 generated samples with the reward oracle, then continue finetuning with SDPO. Differing in our setup from the DNA experiment, we find that using the previously finetuned checkpoint as a reference model during Stage 2 results in superior performance. We provide hyperparameter configurations in Table 6.

Table 6: Detailed hyperparameters for protein inverse folding task.

| Stage | # relabeled samples | $\beta$ | $N$ | # Epochs | Learning Rate |
|---|---|---|---|---|---|
| Stage 1 | 0 | 0.047 | 25 | 9 | 5.7e-6 |
| Stage 2 | 12.8k | 0.063 | 200 | 5 | 8.5e-5 |

---

**Algorithm 1** SDPO Training Algorithm

---

**Input:** Initial dataset $\mathcal{D}^{(0)}$, pretrained discrete diffusion model $p_{\text{ref}}$ and trainable model $p_\theta$.
1: **for** iter $k = 0, \cdots, K$ **do**
2:     **for** step $l = 0, \cdots, L$ **do**
3:         Sample $\{(\mathbf{x}_0^{(i)}, \mathbf{c})\}_{i=1}^N \sim \mathcal{D}^{(k)}$ for each prompt $\mathbf{c}$ in the batch
4:         Sample $\mathbf{x}_t^{(i)} \sim q(\mathbf{x}_t|\mathbf{x}_0)$ given $\mathbf{x}_0^{(i)}$
5:         Compute loss $\mathcal{L}(\theta)$ via Eq. 10             ▷ SDPO loss
6:         $\theta = \theta - \lambda \nabla_\theta \mathcal{L}(\theta)$             ▷ Gradient update
7:     Generate $\mathcal{D}^{(k+1)}$ via $p_\theta(\mathbf{x}_{0:T}|\mathbf{c}) = p(\mathbf{c})p(\mathbf{x}_T) \prod_{t=1}^T p_\theta(\mathbf{x}_{t-1}|\mathbf{x}_t, \mathbf{c})$   ▷ Optional iterative labeling
8:     $p_{\text{ref}} \leftarrow p_\theta$             ▷ Optional reference model update
**Return:** Optimized model $p_\theta$

---

### B.3 LANGUAGE MODELING

We leverage the open-source checkpoint[1] of LLaDA-8B-Instruct (Nie et al., 2025) as the base model to perform our SDPO. We use UltraFeedback (Cui et al., 2023) dataset labeled by Meng et al. (2024) as the finetuning dataset[2]. We operate in the pairwise setting with $N = 2$ on the dataset, with labeled pairs of winning and losing samples with rewards. We use 8 Nvidia 80G A100 GPUs with DeepSpeed enabled during finetuning, due to the scale of the model. We use per device batch size 2 and gradient accumulation of 16 steps, leading to an effective global batch size of 256. We set the learning rate to 1e-6 and $\beta$ to 1.0 and train the model for 2 epochs. At inference time, we reuse the inference hyperparameters adopted in Nie et al. (2025) for GSM8K without any additional tuning, which include total length 256, block size 8, and total number of steps 256. For IFEval and AlpacaEval 2.0, we keep the same set of hyperparameters except setting block size to 32. We always adopt the low confidence remasking strategy, following Nie et al. (2025).

### B.4 COMPLEXITY ANALYSIS

Computational and memory complexity. As an offline preference optimization approach, SDPO is not bottlenecked by online data generation during training, and the offline data generation can be fully parallelized. In detail, it is of $\mathcal{O}(NM(L^2 D + LD^2))$ for computational complexity and $\mathcal{O}(NM(L^2 + LD + D^2))$ for memory complexity, where $N$ is the number of Monte-Carlo samples, $M$ is the number of attention blocks, $L$ is sequence length, and $D$ is the latent dimension. The complexity comes from standard Transformer-based architecture, on top of which the coefficient of is multiplied for Monte-Carlo estimation, making it irrelevant of diffusion steps. The inference complexity remains unaffected.

### B.5 ALGORITHM DETAILS

For completeness, we include the entire training algorithm in Alg. 1. We also include the MC estimator (Alg. 2) to compute the trajectory-level reward $r_t$ used in our reward correlation analysis in ablation study. Note that the function $\text{Constrain}(\mathbf{x}_0^{(i)}, \mathbf{x}_{t-1})$ means setting every unmasked token in $\mathbf{x}_{t-1}$ to the same place in $\mathbf{x}_0^{(i)}$ with the same token value.

---

[1] https://huggingface.co/GSAI-ML/LLaDA-8B-Instruct
[2] https://huggingface.co/datasets/princeton-nlp/llama3-ultrafeedback

---

**Algorithm 2** SDPO Trajectory-level Reward Evaluation

---

**Input:** model $p_\theta$ and $p_{\text{ref}}$, number of MC samples $K$, reward model $r(\cdot)$

1: **for** step $t = T, \cdots, 0$ **do**
2:     Sample $\mathbf{x}_{t-1} \sim p_\theta(\mathbf{x}_{t-1}|\mathbf{x}_t)$
3:     Sample $\{\mathbf{x}_0^{(i)}\}_{i=1}^K$ from $\hat{p}_{\text{ref}}(\mathbf{x}_0|\mathbf{x}_t)$
4:     Estimate the denominator $d = \mathbb{E}_{\hat{p}_{\text{ref}}(\mathbf{x}_0|\mathbf{x}_t)}\left[\exp\left(\frac{1}{\beta_t}r(\mathbf{x}_0,\mathbf{c})\right)\right] \approx \frac{1}{K}\sum_{i=1}^K \exp\left(\frac{1}{\beta_t}r(\mathbf{x}_0^{(i)},\mathbf{c})\right)$
5:     $\tilde{\mathbf{x}}_0^{(i)} \leftarrow \text{Constrain}(\mathbf{x}_0^{(i)}, \mathbf{x}_{t-1})$
6:     Estimate the numerator $n = \mathbb{E}_{p'_{\text{ref}}(\mathbf{x}_0|\mathbf{x}_{t-1},\mathbf{x}_t)}\left[\exp\left(\frac{1}{\beta_t}r(\mathbf{x}_0,\mathbf{c})\right)\right] \approx \frac{1}{K}\sum_{i=1}^K \exp\left(\frac{1}{\beta_t}r(\tilde{\mathbf{x}}_0^{(i)},\mathbf{c})\right)$
7:     Compute the stepwise reward $r_t = \log\frac{n}{d}$

**Return:** chain reward $\hat{r} = \beta \sum_t r_t$

---

Table 7: Dataset size ablation results. Even in highly limited data settings ($< 10\%$ of original dataset), SDPO achieves strong results.

|  | Pred-Activity | ATAC-Acc | 3-mer Corr | App-Log-lik |
|---|---|---|---|---|
| 25k $\rightarrow$ 20k relabeled | 5.56 | 0.40 | 0.795 | -237 |
| 50k $\rightarrow$ 20k relabeled | 6.02 | 0.756 | 0.793 | -248 |
| 700k $\rightarrow$ 20k relabeled | 6.30 | 0.948 | 0.900 | -246 |

Table 8: Detailed results on different values of $\beta$ on DNA design task.

| $\beta$ | Pred-Activity (median) $\uparrow$ | Pred-Activity-std | ATAC-Acc $\uparrow$ (%) |
|---|---|---|---|
| 0.10 | 6.33 | 0.68 | 0.84 |
| 0.15 | 6.29 | 0.68 | 0.85 |
| 0.20 | 6.26 | 0.70 | 0.86 |
| 0.25 | 6.22 | 0.74 | 0.89 |
| 0.30 | 6.17 | 0.74 | 0.91 |
| 0.35 | 6.15 | 0.75 | 0.92 |
| 0.40 | 6.08 | 0.76 | 0.92 |
| 0.45 | 6.01 | 0.85 | 0.92 |
| 0.50 | 6.00 | 0.83 | 0.93 |
| 0.55 | 5.97 | 0.85 | 0.94 |
| 0.60 | 5.90 | 0.84 | 0.94 |
| 0.65 | 5.78 | 0.90 | 0.96 |
| 0.70 | 5.79 | 0.90 | 0.95 |

## C    MORE EXPERIMENT RESULTS

### C.1    ABLATION STUDY ON DATASET SIZE

To study the effect of the data quantity on model performance, we perform an additional ablation on the DNA sequence task in Table 7. Our results demonstrate that SDPO can achieve strong results even in highly limited data settings, where the first stage of fine-tuning uses a small random subset of the original training dataset (700k samples). We follow this with two stages of iterative re-labeling and fine-tuning, according to our established setup.

### C.2    MORE RESULTS ON $\beta$

We provide detailed ablation results on different values of $\beta$ in Table 8 and Table 9 for DNA sequence design and protein inverse folding tasks, respectively. We observe that when $\beta$ becomes smaller, which indicates less regularized distribution *w.r.t.* the reference distribution, the model is granted more flexibility in optimization and generally achieves higher reward. Meanwhile, some other metrics such as ATAC-Acc that relates to the stability of the generated sample will tend to drop due to over-optimizing the model.

Table 9: Detailed results on different values of $\beta$ on protein inverse folding task.

| beta | Pred-ddG (median) ↑ | Pred-ddG-std | %(ddG>0) (%) ↑ |
|------|------|------|------|
| 0.05 | 1.026 | 1.001 | 0.742 |
| 0.10 | 1.119 | 1.093 | 0.760 |
| 0.25 | 0.348 | 1.176 | 0.569 |
| 0.50 | -0.058 | 1.010 | 0.461 |
| 0.75 | -0.350 | 1.215 | 0.430 |
| 1.00 | -0.410 | 1.192 | 0.418 |

## C.3 MORE RESULTS ON PROTEIN INVERSE FOLDING TASK

We provide additional experimental results on the protein inverse folding task, where we demonstrate strong performance in both the re-labeling setting (*a.k.a.* iterative labeling) and the re-labeling-free setting (only using the original dataset for finetuning). The results are depicted in Table 10. In this set of experiments, we obtain the runs that achieve better balance between the target reward (Pred-ddG) and other metrics, such as scRMSD and Success rate. In particular, the experiment with re-labeling achieves better Pred-ddG (1.179) compared with the baseline DRAKES Wang et al. (2024) (1.095) while obtaining better scRMSD and matching its success rate of 78.6%. Furthermore, we observe that without iterative relabeling, SDPO can already obtain strong performance, which additionally underscores the robustness of our design.

Table 10: Additional evaluation on the protein inverse folding task.

| Stage | # relabeled samples | Pred-ddG (median) | %(ddG)> 0 | scRMSD (median) | %(scRMSD)< 2 | Success rate |
|-------|------|------|------|------|------|------|
| w/o Re-labeling | 0 | 1.165 | 85.5 | 0.936 | 90.5 | 77.0 |
| w/ Re-labeling | 12.8k | 1.179 | 84.8 | 0.914 | 92.0 | 78.6 |

## C.4 RESPONSE SAMPLES ON LLaDA-8B-INSTRUCT

Here we present several text samples using the questions from GSM8K Cobbe et al. (2021). It is observed that by employing our SDPO to LLaDA-8B-Instruct, the finetuned model exhibits clearer reasoning logic and better presentation of the response.

---

**Response Samples on GSM8K**

**Question** *[Few-shot Prompts] Janet's ducks lay 16 eggs per day. She eats three for breakfast every morning and bakes muffins for her friends every day with four. She sells the remainder at the farmers' market daily for $2 per fresh duck egg. How much in dollars does she make every day at the farmers' market?*

**Answer by LLaDA-8B-Instruct Nie et al. (2025)**

Janet's ducks lay 16 eggs per day. She eats 3 eggs for breakfast every morning, so she has 16 - 3 = ¡¡16-3=13¿¿13 eggs left. She bakes 4 eggs for her friends every day, so she has 13 - 4 = ¡¡13-4=9¿¿9 eggs left. She sells the remaining eggs at the farmers' market for $2 per egg, so she makes 9 * $2 = $¡¡9*2=18¿¿18 every day.

#### 18

**Answer by LLaDA-8B-Instruct-SDPO (Ours)**

To determine how much Janet makes every day at the farmers' market, we need to follow these steps:

1. Calculate the total number of eggs laid by Janet's ducks per day. 2. Subtract the number of eggs Janet eats for breakfast and bakes for her friends. 3. Calculate the number of eggs left for sale. 4. Multiply the number of eggs left for sale by the price per egg to find her daily earnings.

Let's go through these steps:

1. Janet's ducks lay 16 eggs per day. 2. She eats 3 eggs for breakfast every morning. 3. She bakes 4 eggs for her friends every day.

Now, let's calculate the number of eggs left for sale:

$$16 - 3 - 4 = 9$$

So, Janet has 9 eggs left for sale each day.

Next, we multiply the number of eggs left for sale by the price per egg:

$$9 \times 2 = 18$$

Therefore, Janet makes $18 every day at the farmers' market.

The final answer is $\boxed{18}$

---

**Response Samples on GSM8K**

**Question** *[Few-shot Prompts] Carlos is planting a lemon tree. The tree will cost $90 to plant. Each year it will grow 7 lemons, which he can sell for $1.5 each. It costs $3 a year to water and feed the tree. How many years will it take before he starts earning money on the lemon tree?*

**Answer by LLaDA-8B-Instruct Nie et al. (2025)**

To determine how many years it will take for Carlos to start earning money on the lemon tree, we need to calculate the net profit per year and then see how many years it will take for the net profit to cover the initial cost of planting the tree.

1. **Calculate the net profit per year:**
- Revenue from selling lemons: 7 lemons * $1.5 per lemon = $10.50
- Cost to water and feed the tree: $3
- Net profit per year: $10.50 - $3 = $7.50
2. **Determine the number of years needed to cover the initial cost:**
- Initial cost of planting the tree: $90
- Net profit per year: $7.50
- Number of years needed: $90 / $7.50 per year = 12 years
Therefore, it will take Carlos 12 years to start earning money on the lemon tree.

**Answer by LLaDA-8B-Instruct-SDPO (Ours)**

First, let's calculate the annual revenue from selling the lemons:
- The tree grows 7 lemons each year.
- Each lemon sells for $1.5.
- Annual revenue = 7 lemons * $1.5/lemmon = $10.5.
Next, let's consider the annual cost of watering and feeding the tree:
- It costs $3 per year to water and feed the tree.
Now, let's calculate the net annual profit:
- Net annual profit = Annual revenue - Annual cost

- Net annual profit = $10.5 - $3 = $7.5.
Finally, let's determine how many years it will take for the net annual profit to cover the initial cost of planting the tree:
- Number of years = Initial cost / Net annual profit
- Number of years = $90 / $7.5 = 12 years.
Therefore, it will take Carlos 12 years to start earning money on the lemon tree.

## D    DISCUSSIONS

**Limitation.** Our framework relies on the reward model while, in practice, such model may be noisy or even harmful, which will potentially lead to undesired consequence.

**Broader impact.** As demonstrated in the paper, our approach can help finetune pretrained discrete diffusion models for better alignment towards certain reward, which can have significant practical impact in various domains, such as natural language modeling and biochemical sciences. Our approach can serve as a critical building block towards designing useful DNA and protein sequences, building helpful and harmless chatbots and even performant and effective large language model agentic systems.

## E    THE USE OF LARGE LANGUAGE MODELS

We did not use Large Language Models for research ideation and paper writing in this work.

