# OpenReview forum: "Discrete Diffusion Trajectory Alignment via Stepwise Decomposition"
_ICLR.cc/2026/Conference — ICLR 2026 Poster_

### Official Review · Reviewer_KTkz · 2025-11-01

**Soundness:** 3
**Presentation:** 3
**Contribution:** 2
**Rating:** 4
**Confidence:** 5

**Summary:**

This paper proposes an offline fine-tuning method for discrete diffusion, without rely on the reward gradient calculations.

**Strengths:**

The overall framework and results are technical sound. The proposed fine tuning methods without gradient calculation are important in scientific area.

**Weaknesses:**

- Performances: in Table 1, it’s reported that DRAKES achieve Pred-Activity as 5.61 while proposed SDPO is 6.30. However, in the original paper, DRAKES have two versions (with and without KL), which is 5.61 and 6.44, respectively. Why the authors only pick up the lower value one? If DRAKES can achieve 6.44, does it mean the performance gain by SDPO is low? Since the primary optimization reward here is Pred-Activity, I think this is a big experimental flaw.
- Baselines: authors may miss some important baselines. For fine-tuning discrete diffusion without gradient, DDPP [1] and DDPO [2] serve as strong baselines. Why not add them as the baseline methods?

[1] Rector-Brooks J, Hasan M, Peng Z, et al. Steering masked discrete diffusion models via discrete denoising posterior prediction[J]. arXiv preprint arXiv:2410.08134, 2024.
[2] Black K, Janner M, Du Y, et al. Training diffusion models with reinforcement learning[J]. arXiv preprint arXiv:2305.13301, 2023.

**Questions:**

- In Line 262-263, the authors show that $x_t$ cannot be calculated in offline setting, so that the authors drop it. Although it’s understandable, could the authors provide more in-depth analysis on this? Will this cause big problems?
- In iterative labeling, authors present that they will update the datasets from latest models. Does it mean that the overall framework is not completely offline anymore?
- While I appreciate the authors add diversity quantity in fine-tuning diffusions, in Table 2, why the pretrained model doesn’t have the highest entropy? My understanding is that, no matter how you can keep diversity, during fine-tuning of reward maximization, adding diversity measure from pretrained model of 35.2 to 42.3 seems strange to me.
- One important measurement is the computation cost. Compared with your baselines, did your methods add additional computation costs, especially the reward calculations? (since scientific reward calculation may be very cost) How about the convergence rate under same computation compared with the baselines?

---

> ### Author Response · Authors · 2025-11-20
>
> Thank you for the detailed review and suggestions! We provide point-to-point response to your comments as follows.
>
> > **W1. Performances: in Table 1, it’s reported that DRAKES achieve Pred-Activity as 5.61 while proposed SDPO is 6.30. However, in the original paper, DRAKES have two versions (with and without KL), which is 5.61 and 6.44, respectively. Why the authors only pick up the lower value one? If DRAKES can achieve 6.44, does it mean the performance gain by SDPO is low?**
>
>
> Thanks for pointing this out! In the trajectory alignment setting, we prioritize ability not only to optimize for reward value, but also to prevent over-fitting to the reward model. For this reason, we did not include DRAKES without KL, because despite its strong reward optimization capability, it clearly shows deterioriation in other metrics. We would like to assure the reviewer that SDPO is also quite capable of reward optimization; we have various runs of our method that significantly outperform DRAKES without KL in Pred-HepG2, but opted to present our current table's results because it provided the strongest balance between reward optimization and preservation of sample naturalness. For example, we have a different run of SDPO that achieves the following:
>
> ||Pred-HepG2|3-mer Pearson Corr.|ATAC Accuracy|JASPAR|Approximate Log-Likelihood|
> |-|-|-|-|-|-|
> |SDPO|6.77| 0.872|0.814|0.933|-243|
> |DRAKES w/o KL|6.44|0.307|0.825|0.557|-281
>
> These results strictly outperform DRAKES without KL on nearly all metrics, showing a significant improvement in Pred-HepG2, while also maintaining naturalness of samples, as demonstrated by strong performance in other metrics.
>
> Moreover, for the protein task, DRAKES w/o KL reports 1.108 Pred-ddG, while our SDPO achieves a much higher Pred-ddG of 1.400, while maintaining significantly better naturalness as evidenced by the other metrics. Overall, SDPO is consistently more performant than DRAKES for reward optimization and maintaining naturalness of samples.
>
>
> > **W2. Baselines: authors may miss some important baselines. For fine-tuning discrete diffusion without gradient, DDPP [1] and DDPO [2] serve as strong baselines. Why not add them as the baseline methods?**
>
> Thank you for pointing out the additional baselines. In particular, DDPO [2] is a policy gradient-based approach for continuous Gaussian diffusion and we instead include diffu-GRPO [3] a more advanced policy gradient approach tailored for discrete diffusion.
> We also evaluated DDPP-IS [1], an importance sampling method to match the reward-tilted posterior, on the DNA task.
>
> ||Pred-HepG2|3-mer Pearson Corr.|ATAC Accuracy|JASPAR|Approximate Log-Likelihood|
> |-|-|-|-|-|-|
> |diffu-GRPO|5.86|0.783|33.0|0.903|-245|
> |DDPP-IS|4.07|0.711|0.50|0.723|-253|
> |SDPO|**6.30**|**0.900**|**0.948**|**0.936**|-246|
>
> We also compare with diffu-GRPO on the language modeling task:
>
>
> | |AlpacaEval 2.0 LC|AlpacaEval 2.0 WR|GSM8K|IFEval|
> |-|-|-|-|-|
> |LLaDA-8B-Instruct|10.6|6.8|78.6|52.9|
> |w/ diffu-GRPO|12.6|7.8|80.5|53.5|
> |w/ SDPO|**14.2**|**8.7**|**81.2**|**55.1**|
>
> The results show that our SDPO consistently outperform the policy gradient-based approach diffu-GRPO and the posterior matching approach DDPP-IS. We have also updated the manuscript to include these baselines.
>
>
> [1] Rector-Brooks et al. Steering masked discrete diffusion models via discrete denoising posterior prediction.
>
> [2] Black et al. Training diffusion models with reinforcement learning.
>
> [3] Zhao et al. d1: Scaling Reasoning in Diffusion Large Language Models via Reinforcement Learning.
>
> > **Q1. In Line 262-263, the authors show that $x_t$ cannot be calculated in offline setting, so that the authors drop it. Although it’s understandable, could the authors provide more in-depth analysis on this? Will this cause big problems?**
>
> In Equation 9, the Monte-Carlo estimation of the partition $Z\_\theta^t$ requires to sample pairs of $\mathbf{x}\_0$ and $\mathbf{x}\_t$ from the posterior $p\_\mathrm{ref}(\mathbf{x}\_0|\mathbf{x}\_t)$. Since $\mathbf{x}\_t$ cannot be directly obtained, we instead sample from the joint $p\_\text{ref}(\mathbf{x}\_0,\mathbf{x}\_t)$, which can be approximated by the forward process by first sampling $\mathbf{x}\_0$ from data and then $\mathbf{x}\_t$ from $q(\mathbf{x}\_t|\mathbf{x}\_0)$. Since the reference model $p\_\text{ref}$ is a pretrained discrete diffusion model, the error in this approximation has already been sufficiently reduced in the pretraining stage through optimizing the diffusion loss. Notably, this approximation has been similarly leveraged in previous diffusion preference optimization methods [1,2], and the strong empirical performance of SDPO demonstrated in our experiments also verifies the validity of our design here.
>
> [1] Wallace et al. Diffusion Model Alignment Using Direct Preference Optimization.
>
> [2] Borso et al. Preference-Based Alignment of Discrete Diffusion Models.

---

> > ### Author Response · Authors · 2025-11-20
> >
> > > **Q2. In iterative labeling, authors present that they will update the datasets from latest models. Does it mean that the overall framework is not completely offline anymore?**
> >
> > Yes. If we enable iterative labeling, the framework will be slightly relaxed from a strict offline setting. This is beneficial in the case where the pretrained dataset itself contains a significant portion of low reward samples and we may need to update the dataset with samples of higher reward by sampling from the finetuned model to further enhance the performance.
> >
> > Notably, in our ablation study for iterative labeling, we have shown that with only a small amount of additional samples, we can significantly improve the performance, which demonstrates the data-efficiency of enabling iterative labeling in our approach. In Table 1's results for the DNA task, SDPO achieves superior performance with only 20k additional samples, much less than the 128k samples used by DRAKES and diffu-GRPO.
> >
> >
> > > **Q3. While I appreciate the authors add diversity quantity in fine-tuning diffusions, in Table 2, why the pretrained model doesn’t have the highest entropy?**
> >
> > Thank you for raising this point. While reward optimization does often reduce diversity of generated samples, this is not always the case. For the protein task, the pre-trained diffusion model places mass in a relatively low-reward region of sequence space, whereas fine-tuning shifts probability toward a high-reward region identified by the reward model. This shift can increase diversity if the high-reward region is itself diverse. We hypothesize that this induced exploration of the sample space leads to greater diversity for the protein task; in fact, several other baselines, such as D2-DPO and VRPO, also have increased average entropy when compared to the pre-trained model.
> >
> >
> > > **Q4. Compared with your baselines, did your methods add additional computation costs, especially the reward calculations? (since scientific reward calculation may be very cost) How about the convergence rate under same computation compared with the baselines?**
> >
> > This is an interesting point! Our SDPO is in fact the most computationally efficient method in terms of number of reward calculations, training efficiency, and convergence rate, as we will detail below.
> >
> > **Reward calculations.** We compare the total number of reward calculations of SDPO and DRAKES in the table below.
> >
> > ||DNA Design|Protein Inverse Folding|
> > |-|-|-|
> > |DRAKES|128k|12.8k|
> > |SDPO|20k|10k|
> >
> > It is clearly observed that SDPO uses much fewer reward calculations than DRAKES while achieving better overall performance, which further demonstrates the efficacy of our approach.
> >
> > **Training efficiency.** We further compare the actual wallclock time per training step between SDPO and the baselines DRAKES and diffu-GRPO.
> >
> > DNA Task:
> > ||DRAKES|diffu-GRPO|SDPO|
> > |-|-|-|-|
> > |Time per Step (sec)|6.02|1.51|0.77|
> >
> > Language modeling Task on LLaDA-8B-Instruct:
> > ||diffu-GRPO|SDPO|
> > |-|-|-|
> > |Time per Step (sec)|52.46|21.11|
> >
> >
> > Notably, without the need to perform online sampling, SDPO is near 7.8$\times$ more efficient than DRAKES and 2.2$\times$ more efficient than diffu-GRPO in terms of wallclock time per training step, while achieving higher rewards and better overall performance.
> >
> > **Convergence rate.** We also perform a systematic head-to-head comparison of the reward convergence rate in **Fig. 4(a)** in the ablation study section of the updated paper. Specifically, our SDPO with iterative labeling achieves 6.2 Pred-HepG2 using only **15k** labeled samples, while DRAKES and diffu-GRPO only achieve 5.6 and 4.2 Pred-HepG2 with **25k** labeled samples.
> >
> > Putting together, we highlight that SDPO is indeed much more efficient in both training speed and convergence rate, making it particularly favorable compared with the baselines. We have also updated the paper to include these important discussions and we sincerely thank you for bringing up this point!
> >
> >
> > **We sincerely hope our response could address your concerns!**

---

> > ### Author Response · Authors · 2025-11-29
> >
> > We also performed evaluation of the baselines diffu-GRPO and DDPD-IS on the protein inverse folding task:
> >
> > ||Pred-ddG|%(ddG>0)|scRMSD|%(scRMSD<2)|Success Rate| Entropy|
> > |-|-|-|-|-|-|-|
> > |DDPD-IS|-0.130|46.7| 0.829|89.3|43.3| 24.3 |
> > |diffu-GRPO|1.286|76.8| 1.192|57.1|37.2| 40.0 |
> > |SDPO|**1.400**|**87.1**| 0.938|88.9|**75.5**| **42.3** |
> >
> > Clearly, SDPO yields higher reward while maintaining significantly higher success rate and lower scRMSD, while diffu-GRPO leads to notable quality degradation and DDPD-IS results in less effective reward optimization. The result again verify the superiority of our approach.
> >
> > **We sincerely hope our response could address your concerns!**

---

### Official Review · Reviewer_pkbk · 2025-11-01

**Soundness:** 2
**Presentation:** 3
**Contribution:** 2
**Rating:** 4
**Confidence:** 4

**Summary:**

The paper propose a preference optimization method for discrete diffusion models using diffusion trajectory alignment. It decompose the alignment problem into a set of stepwise objectives, instead of directly optimization for the reward on the final output. The authors provide the optimality guarantee under the assumption of the additive factorization of the trajectory reward. Experiments on DNA sequence, protein sequence, and language demonstrate its effectiveness.

**Strengths:**

- Experiments on different domains demonstrate strong performance across different metrics.
- The idea to leverage the intermediate step information for alignment of discrete diffusion models is well-motivated.

**Weaknesses:**

- Although claimed as a trajectory alignment algorithm to maximize the stepwise reward, the algorithm does not really use the stepwise reward. Instead, it only uses the reward of the clean sample $r(x\_0,c)$ . Actually, the trajectory alignment here is more like a simple decomposition of the joint distribution for the data itself, while irrelevant to the reward-tilted part.
- In Theorem 4.1, the optimality of the trajectory alignment objective is achieved when the reward $\hat{r}(x_{0:T},c)$ is defined as the complicated chain reward formulation. Therefore, the theorem actually defines a very strong condition specialized for the trajectory alignment objective to prove that the trajectory alignment objective is optimal under this specialized condition. Although we may gain some interesting insights from the condition, the theorem itself is not very meaningful.
- The final objective relies on the coefficient w(t). Could the author provide more ablations on different choices of w(t) to better demonstrate its effect?
- The authors turn this trajectory alignment objective into an off-policy preference optimization objective. For general RL algorithms of LLMs, it is known that on-policy RL works better than off-policy RL methods. Why does the author choose off-policy preference optimization as the final objective, instead of an on-policy one, and does SDPO suffer from the same issue as off-policy RL for LLMs?

**Questions:**

Please refer to the **weaknesses** section.

---

> ### Author Response · Authors · 2025-11-20
>
> Thank you for the detailed review and suggestions! We provide point-to-point response to your comments as follows.
>
> > **W1. Although claimed as a trajectory alignment algorithm to maximize the stepwise reward, the algorithm does not really use the stepwise reward. Instead, it only uses the reward of the clean sample $r(x_0,c)$.**
>
> In the problem of diffusion model alignment, practically we only have access to the reward model that operates on the clean sample, and there is **no** "stepwise" reward model readily available. Therefore, it is not a weakness of not explicitly using any stepwise reward model, but instead an advantage.
>
> Moreover, our SDPO introduces a stepwise alignment objective (Equation 7) that, critically, **induces a stepwise reward**  (i.e., $r\_t(\mathbf{x}\_{t-1};\mathbf{x}\_t,\mathbf{c})$ in Theorem 4.1). This indicates that optimizing our SDPO loss is equivalent to training the diffusion chain with the induced stepwise reward $r\_t(\mathbf{x}\_{t-1};\mathbf{x}\_t,\mathbf{c})$, which adds dense and exact supervision to diffusion alignment. Therefore, we can effectively perform diffusion alignment with the induced stepwise reward without the need to explicitly train a stepwise reward model.
>
>
> > **W2. Theorem 4.1 actually defines a very strong condition specialized for the trajectory alignment objective to prove that the trajectory alignment objective is optimal under this specialized condition. Although we may gain some interesting insights from the condition, the theorem itself is not very meaningful.**
>
> Thank you for the comment. However, we would like to respectfully point out that Theorem 4.1 is indeed central to our analysis and crucial to justify the utility of our approach. In fact, Theorem 4.1 conveys the following information: If we align the posterior $p\_\theta(\mathbf{x}\_0|\mathbf{x}\_t)$ for each step $t$ according to Equation 7, we are equivalently optimizing the diffusion model with the stepwise reward $r\_t(\mathbf{x}\_{t-1};\mathbf{x}\_t,\mathbf{c})$, such that the cumulative reward of a reverse sampling chain will be $\hat{r}(\mathbf{x}\_{0:T},\mathbf{c})=\beta\sum\_t r\_t(\mathbf{x}\_{t-1};\mathbf{x}\_t,\mathbf{c})$, and correspondingly the optimized reverse sampling distribution will be $p^\ast(\mathbf{x}\_{0:T}|\mathbf{c})\propto p\_\text{ref}(\mathbf{x}\_{0:T}|\mathbf{c})\exp(\hat{r}(\mathbf{x}\_{0:T},\mathbf{c})/\beta)$.
>
> The theorem is indeed very useful in the following senses:
>
> * First, it gives the **exact** expression of the reverse joint $p^\ast(\mathbf{x}\_{0:T}|\mathbf{c})$ under the stepwise alignment objective (Equation 7). Therefore, it precisely characterizes the distribution of the samples obtained from the reverse sampling process, when the model is sufficiently finetuned with the stepwise objective. In contrast, previous works for diffusion alignment only give a bound [1] or the gradient analysis of the loss function [2], but **not** an **exact expression** of the reverse sampling distribution.
> * Second, the theorem unveils an important insight: optimizing the stepwise objective in Equation 7 essentially induces a stepwise reward model $r\_t(\mathbf{x}\_{t-1};\mathbf{x}\_t,\mathbf{c})$. This observation justifies the validity of our approach: optimizing the stepwise alignment objective increases the stepwise reward $r_t$, which consequently promotes the trajectory-level reward $\hat{r}(\mathbf{x}\_{0:T},\mathbf{c})$. Notably, we have also performed a correlation analysis between the trajectory-level reward $\hat{r}(\mathbf{x}\_{0:T},\mathbf{c})$ and the original reward $r(\mathbf{x}\_0,\mathbf{c})$ and found a strong correlation between them (**Fig. 4(b)** in ablation study). Therefore, by training the model to increase $\hat{r}(\mathbf{x}\_{0:T},\mathbf{c})$, the final reverse sampling chain will lead to clean sample $\mathbf{x}\_0$ with high reward $r(\mathbf{x}\_0,\mathbf{c})$ as desired.
>
> Our experiments also support this finding that with the dense stepwise reward signal, our SDPO achieves much higher convergence speed while yielding superior sample reward, which also mirrors the findings in Theorem 4.1.
>
> [1] Wallace et al. Diffusion Model Alignment Using Direct Preference Optimization.
>
> [2] Zhu et al. DSPO: Direct Score Preference Optimization for Diffusion Model Alignment.

---

> > ### Author Response · Authors · 2025-11-20
> >
> > > **W3. The final objective relies on the coefficient w(t). Could the author provide more ablations on different choices of w(t) to better demonstrate its effect?**
> >
> > Thank you for raising this point! We have performed additional ablation studies on the choice of $w(t)$ in  the language modeling task, where we investigate three different choices of $w(t)$, namely $w(t)=1-\alpha_t$, $w(t)=1$, and $w(t)=\alpha_t$. The results are depicted below:
> >
> > |$w(t)$ |AlpacaEval 2.0 LC|AlpacaEval 2.0 WR|GSM8K|IFEval|
> > |-|-|-|-|-|
> > |$\alpha_t$|13.2|8.2|80.5|54.8|
> > |1|13.8|8.4|80.8|54.6|
> > |$1-\alpha_t$|**14.2**|**8.7**|**81.2**|**55.1**|
> >
> > We observe that using $w(t)=1-\alpha_t$ overall leads to the best performance while other schedulers of $w(t)$ lead to a minor drop in the metrics. We speculate that when $w(t)=1-\alpha_t$ the SDPO loss is effectively amortized to each masked token (since $1-\alpha_t$ is the mask rate at step $t$) which leads to lower variance in the gradient.
> >
> >
> > > **W4. Why does the author choose off-policy preference optimization as the final objective, instead of an on-policy one, and does SDPO suffer from the same issue as off-policy RL for LLMs?**
> >
> > Thank you for the interesting question! Compared with on-policy method, off-policy approach can directly reuse the pretraining dataset and does not require costly online sampling for each training iteration, thus being more training-efficient. To verify the training efficiency of our SDPO, we compare the average wallclock time per iteration on DNA and language modeling tasks:
> >
> > DNA Task:
> > ||DRAKES|diffu-GRPO|SDPO|
> > |-|-|-|-|
> > |Time per Step (sec)|6.02|1.51|0.77|
> >
> > Language modeling Task on LLaDA-8B-Instruct:
> > ||diffu-GRPO|SDPO|
> > |-|-|-|
> > |Time per Step (sec)|52.46|21.11|
> >
> > Notably, without the need to perform online sampling, SDPO is near 7.8$\times$ more efficient than DRAKES and 2.2$\times$ more efficient than diffu-GRPO in terms of wallclock time per training step, while achieving higher rewards and better overall performance.
> >
> > However, as the reviewer has pointed out, off-policy algorithms also have limitations. Notably, off-policy methods fully rely on the offline dataset without performing sampling from the current model, which constrains exploration and usually leads to insufficient reward enhancement.
> >
> > To tackle this challenge, we have introduced a variant of SDPO, namely SDPO with iterative labeling, that effectively brings the on-policy flavor into the algorithm. Specifically, we re-generate a set of samples from the current optimized model and label them with the reward model, and use this new dataset to perform a new round of SDPO training. Our best results in the DNA and protein tasks were indeed obtained using this iterative labeling technique, and we have also systematically investigated its effect in the ablation study section. More importantly, we are able to obtain the best results with only a very small amount of additional samples, compared with DRAKES and diffu-GRPO, which desmontrates the superior reward efficiency of SDPO with iterative labeling.
> >
> > To offer more insights, we also add an additional head-to-head comparison of SDPO and the iterative labeling variant in **Fig. 4(a)** in the revised manuscript. The curve shows that SDPO is more reward-efficient and converges much faster compared with DRAKES and diffu-GRPO, while the iterative labeling variant of SDPO (SDPO w/ Full Iter) achieves significantly higher reward, validating our design in relaxing SDPO with on-policy training using iterative labeling.
> >
> >
> > **We sincerely hope our response could address your concerns!**

---

### Official Review · Reviewer_Gnse · 2025-11-01

**Soundness:** 2
**Presentation:** 3
**Contribution:** 3
**Rating:** 4
**Confidence:** 4

**Summary:**

The work aims to fine-tune a discrete diffusion model (with trajectory-level distribution $p_\theta(x_{0:T})$) to sample from the tilted distribution  $p_{\theta}(x_{0:T}) \exp(\frac{1}{\beta}r(x_{0:T}))$. It proposes a surrogate objective which decomposes the trajectory-level problem into subproblems where each posterior over $x_0$: $p_\theta(x_0|x_t)$ is fine-tuned to the target $p_\theta(x_0 | x_t) \exp(\frac{1}{\beta_t}r(x_0))$, which side-steps issues with expensive likelihood evaluations over the trajectory. The work focuses on the offline-setting where a reward-labeled dataset (with data drawn from $p_\theta$) is available, and solves each subproblem using a variant of forward-KL/cross-entropy loss, where samples from the data-set are importance weighted using the reward. The method is shown to outperform other baselines on the tasks of DNA sequence design, protein inverse folding, and preference alignment for language modelling.

**Strengths:**

1. Side-stepping the trajectory-level nature of discrete diffusion fine-tuning is an important problem, and the proposed method is a useful step in this direction.
2. The work is well organized.
3. The preference alignment task in language modelling demonstrates scalability of the method to larger models.
4. The extension to iterative training is useful.
5. The ablations section is well-done, and clarifies important aspects of the method (namely its sensitivity to the sample size $N$).
6. The evaluations are done on a comprehensive set of metrics for DNA sequence design and protein inverse folding.

**Weaknesses:**

1. The method assumes access to $p_\theta(x_0|x_t)$ - however, in discrete diffusion, we only have access to a factorized approximation (or equivalently, the **mean**) (eg. see section 3 of (Shi et al., 2024)) of this posterior (through $f_\theta(x_t,t)$). Equation 12 conflates these two things as being equal.
    - This additionally calls into question the applicability of the theoretical analysis in Theorem 4.1 - does the result still hold for the factorized approximation of the posterior $p_\theta(x_0 | x_t)$?
2. The variance of the importance weights (or equivalently the effective sample size) should be evaluated for various tasks - since a reduced sample size appears to be a potential limitation of the method for more challenging rewards.
3. The bias in using the stepwise approximation should be better analyzed:
    - Are there real rewards for which the method is unbiased (at sampling from the trajectory level reward-tilted distribution)?
    - If not, a visualization of how the method is biased would clarify the methods drawbacks (for instance on a toy task where the exact reward-tilted posterior can be visualized).

4. The choice of baselines lack a more standard policy gradient RL method. I understand that the method tackles an offline setting, but I think at least one comparison to such a method (such as diffu-GRPO (Zhao et al., 2025)) would be helpful in understanding the sample efficiency of the proposed method. In particular this is because such methods have been shown to be more scalable to challenging problems than DRAKES.

5. The work is in need of proofreading for various grammatical mistakes.

For now I am recommending a weak reject, mainly due to point 1 and its impact on the theoretical results (and secondarily, point 4 as an important missing baseline) above.

Jiaxin Shi, Kehang Han, Zhe Wang, Arnaud Doucet, and Michalis K Titsias. Simplified and
generalized masked diffusion for discrete data. arXiv preprint arXiv:2406.04329, 2024.

Siyan Zhao, Devaansh Gupta, Qinqing Zheng, and Aditya Grover. d1: Scaling reasoning in diffusion
large language models via reinforcement learning. 2025. arXiv:2504.12216.

**Questions:**

1. Why not formulate Equation (8) as $D_{\mathrm{KL}}( p_\mathrm{ref}(x_0 | x_t, c) \exp(r(x_0, c)/\beta_t) \mid \mid p_\theta(x_0 | x_t, c))$ directly? Is there an advantage to the loss as its presented?

2. In Appendix B, it is mentioned that the first stage of training is done on existing/original datasets for fine-tuning. This in theory creates a mismatch with the training objective since it assumes the pretrained model’s distribution perfectly coincides with its pretraining dataset, which is likely not true. Does the performance of the method degrade if samples from the pretrained model are used instead, for the initial fine-tuning stage?

---

> ### Author Response · Authors · 2025-11-20
>
> Thank you for the detailed review and suggestions! We provide point-to-point response to your comments as follows.
>
>
> > **W1. The method assumes access to $p\_\theta(x\_0|x\_t)$ - however, in discrete diffusion, we only have access to a factorized approximation of this posterior. Equation 12 conflates these two things as being equal.**
>
> Thank you for the insightful comment! In fact, all $p\_\theta(\mathbf{x}\_0|\mathbf{x}\_t)$ we use throughout the paper exactly refers to the factorized approximation, following [1,2].
> Therefore, Equation 12 is correct since $p_\theta(\mathbf{x}_0|\mathbf{x}_t)$ is precisely parameterized in this form. Notably, the proof of Theorem 4.1 also works under the factorized approximation, as we will explain in the answer to W1.1 below.
>
> With the clarifications, we fully agree with the reviewer that this point may cause confusion if it is not clearly stated. We have therefore updated the manuscript to **1.** change the notation from $p\_\theta(\mathbf{x}\_0|\mathbf{x}\_t)$ to $\hat{p}\_\theta(\mathbf{x}\_0|\mathbf{x}\_t)$, and **2.** explicitly define $\hat{p}\_\theta(\mathbf{x}\_0|\mathbf{x}\_t)=\prod_i\hat{p}_\theta(\mathbf{x}_0^{(i)}|\mathbf{x}_t)$ ($i$ is the token index in the block)  as the factorized form, to explicitly show that the posterior we use is the factorized approximation.
>
> Please refer to the updated version for the detailed modifications.
>
>
>
> > **W1.1. This additionally calls into question the applicability of the theoretical analysis in Theorem 4.1 - does the result still hold for the factorized approximation of the posterior $p\_\theta(x\_0|x\_t)$?**
>
> Yes, Theorem 4.1 still holds under the factorized approximation. The core step in our proof of Theorem 4.1 that links the stepwise alignment $\hat{p}\_\theta(\mathbf{x}\_0|\mathbf{x}\_t)$ to the reverse joint $p\_\theta(\mathbf{x}\_{0:T})$ is the $\mathbf{x}\_0$-parameterization of the reverse transition kernel $p\_\theta(\mathbf{x}\_{t-1}|\mathbf{x}\_t)=\sum\_{\mathbf{x}\_0} \hat{p}\_\theta(\mathbf{x}\_0|\mathbf{x}\_t)q(\mathbf{x}\_{t-1}|\mathbf{x}\_t,\mathbf{x}\_0)$ (c.f. Equation 4 in [2]), where $\hat{p}\_\theta(\mathbf{x}\_0|\mathbf{x}\_t)$ is exactly the factorized approximation of the posterior. We have also updated the notation in the proof of Theorem 4.1 to $\hat{p}\_\theta(\mathbf{x}\_0|\mathbf{x}\_t)$ to avoid confusion.
>
>
> [1] Shi et al. Simplified and Generalized Masked Diffusion for Discrete Data.
>
> [2] Austin et al. Structured Denoising Diffusion Models in Discrete State-Spaces.
>
>
> > **W2. The variance of the importance weights (or equivalently the effective sample size) should be evaluated for various tasks.**
>
> Thank you for the interesting point! During training, we form importance weights $w_i \propto \exp(r(x_0^{(i)},c))$. We have computed the average variance of importance weights and effective sample size (ESS = $1 / \sum_i w_i^2$) per group of $N$ samples, for different values of $N$ in both the DNA and protein tasks.
>
> |DNA|Average importance weight variance|Average ESS|
> |-|-|-|
> |$N=2$|0.113|1.69|
> |$N=10$|0.0228|4.76|
> |$N=25$|0.00869|7.22|
> |$N=50$|0.0039|9.09|
>
> |Protein|Average importance weight variance|Average ESS|
> |-|-|-|
> |$N=2$|0.184|1.54|
> |$N=10$|0.00929|5.73|
> |$N=25$|0.00156|13.47|
> |$N=50$|0.000403|26.05|
>
> As seen for both tasks, a larger $N$ leads to larger ESS and lower variance in importance weights. We emphasize that SDPO remains robust, empirically demonstrating strong performance across multiple values of $N$ in the DNA and protein tasks.

---

> > ### Author Response · Authors · 2025-11-20
> >
> > > **W3. The bias in using the stepwise approximation should be better analyzed.**
> >
> > Thank you for the advice! The optimal of our SDPO is given by $p^\beta\_{\hat{r}}(\mathbf{x}\_{0:T}|\mathbf{c})\propto p\_\text{ref}(\mathbf{x}\_{0:T}|\mathbf{c})\exp(\hat{r}(\mathbf{x}\_{0:T},\mathbf{c})/\beta)$ where $\hat{r}$ is the chain reward defined in Theorem 4.1, while the typical optimal clean reward-tilted distribution is given by $p^\beta\_r(\mathbf{x}\_{0:T}|\mathbf{c})\propto p\_\text{ref}(\mathbf{x}\_{0:T}|\mathbf{c})\exp(r(\mathbf{x}\_{0},\mathbf{c})/\beta)$. We fully agree that deeper investigations on the relationship between our $\hat{r}(\mathbf{x}\_{0:T},\mathbf{c})$ and the original $r(\mathbf{x}\_0,\mathbf{c})$ will help gain more insights in the validity and potential bias introduced in the algorithm.
> >
> > We have therefore conducted the following experiment on the DNA task to investigate the relationship between the rewards $\hat{r}$ and $r$, which equivalently implies the relationship between our reward-tilted distribution $p^\ast\_{\hat{r}}$ and the exact reward-tilted distribution $p^\ast\_r$. Specifically, we sample 50 trajectories from the pretrained model, the model after first stage training, and the final model respectively, leading to 150 trajectories in total $\{\mathbf{x}^{(i)}\_{0:T}\}\_{i=1}^{150}$. For each trajectory $\mathbf{x}^{(i)}\_{0:T}$, we evaluate its chain reward $\hat{r}^{(i)}=\hat{r}(\mathbf{x}^{(i)}\_{0:T},\mathbf{c})$ using an unbiased MC estimator (see Appendix B.5 for more details) and its clean reward $r^{(i)}=r(\mathbf{x}^{(i)}\_0,\mathbf{c})$. We then perform linear correlation analysis  for the set of datapoints $\{( r^{(i)},\hat{r}^{(i)} ) \}\_{i=1}^{150}$. We provide the scatter plot of the linear regression in **Fig.4(b)** of the updated manuscript.
> >
> > In particular, despite using MC estimation, we observe a relatively strong positive correlation between the chain reward and the clean reward, with a Pearson correlation of 0.781. This indicates that the chain reward can be approximately viewed as $\hat{r}\approx a\cdot r + b$, which draws an interesting connection between the reward-tilted distributions: $p^\beta\_{\hat{r}}(\mathbf{x}\_{0:T}|\mathbf{c})\propto p\_\text{ref}(\mathbf{x}\_{0:T}|\mathbf{c})\exp(\hat{r}(\mathbf{x}\_{0:T},\mathbf{c})/\beta)\approx p\_\text{ref}(\mathbf{x}\_{0:T}|\mathbf{c})\exp(a\cdot r(\mathbf{x}\_0,\mathbf{c})/\beta + b/\beta)$
> > $\propto p\_\text{ref}(\mathbf{x}\_{0:T}|\mathbf{c})\exp(a\cdot r(\mathbf{x}\_0,\mathbf{c})/\beta) \propto p^{\beta/a}\_r(\mathbf{x}\_{0:T}|\mathbf{c})$
> >
> > Therefore, this empirical investigation interestingly reveals that our chain reward is an effective surrogate of the clean reward and, more importantly, our optimal reward-tilted distribution is also a decent approximation of the original trajectory-level clean reward-tilted distribution without notable bias introduced.
> >
> > We have also added this analysis to the ablation study section of the revised paper.
> >
> > > **W4. The choice of baselines lack a more standard policy gradient RL method. I understand that the method tackles an offline setting, but I think at least one comparison to such a method (such as diffu-GRPO (Zhao et al., 2025)) would be helpful.**
> >
> > Thank you for the advice. We have performed additional evaluation of diffu-GRPO on the DNA sequence task, achieving the following results:
> >
> > ||Pred-HepG2|ATAC Accuracy|3-mer Pearson Corr.|JASPAR|Approximate Log-Likelihood| Time per Step (sec)|
> > |-|-|-|-|-|-|-|
> > |diffu-GRPO|5.86|33.0| 0.783|0.903|-245| 1.51 |
> > |SDPO|**6.30**|**94.8**| **0.900**|**0.936**|-246| **0.77** |
> >
> > We follow the same compute budget as DRAKES, using a total of 128k generated samples/reward model calls. Notably, SDPO outperforms diffu-GRPO on all metrics, while also permitting significantly faster training speed since no online sampling is required for each iteration.
> >
> > Besides the DNA sequence design task, we also perform additional comparison with diffu-GRPO in our language modeling experiments. We follow the benchmark protocol depicted in Sec. 5.3 to finetune LLaDA-8B-Instruct and the results are presented below:
> >
> > | |AlpacaEval 2.0 LC|AlpacaEval 2.0 WR|GSM8K|IFEval|Time per Step (sec)|
> > |-|-|-|-|-|-|
> > |LLaDA-8B-Instruct|10.6|6.8|78.6|52.9|-|
> > |w/ diffu-GRPO|12.6|7.8|80.5|53.5|52.46|
> > |w/ SDPO|**14.2**|**8.7**|**81.2**|**55.1**|**21.11**|
> >
> > Similar to the observations on DNA task, SDPO achieves better performance consistently across three benchmarks, while being 2.5$\times$ faster in training for each step, compared with the online RL method diffu-GRPO. Moreover, SDPO only took 8 hours in total to achieve the performance reported above, while diffu-GRPO training took more than 30 hours, yet still inferior to the performance of SDPO. The results further verify the efficacy of our SDPO in terms of improving model performance with significantly higher training efficiency.

---

> ### Author Response · Authors · 2025-11-20
>
> > **W5. The work is in need of proofreading for various grammatical mistakes.**
>
> Thank you. We have gone through a detailed proofreading and corrected the grammatical mistakes.
>
> > **Q1. Why not formulate Equation (8) as $D\_\mathrm{KL}(p\_\text{ref}(x\_0|x\_t,c)\exp(r(x\_0,c)/\beta\_t)\|p\_\theta(x\_0|x\_t,c))$ directly? Is there an advantage to the loss as its presented?**
>
> This is an interesting point since both the formulation you proposed $D\_\mathrm{KL}[p\_\text{ref}(x\_0|x\_t,c)\exp(r(x\_0,c)/\beta\_t)\|p\_\theta(x\_0|x\_t,c)]$ (\*) and our proposal in Equation 8 lead to the same optima $p\_\theta(x\_0|x\_t,c)\propto p\_\text{ref}(x\_0|x\_t,c)\exp(r(x\_0,c)/\beta\_t)$. However, they result in different empirical loss functions and hence different optimization. In fact, (\*) will arrive at the following loss function, mirroring our derivations from Equation 8 to 9:
> $\mathcal{L}'\_t(\theta)=-\mathbb{E}\_{c,x\_0,x\_t}\left[\frac{\exp(r(x\_0,c)/\beta\_t)}{Z\_r(c,\beta\_t)} \log \frac{\exp(r\_\theta(x\_0,x\_t,c))}{Z_\\theta^t(x\_t,c)}\right],$
> where $r_\theta(x_0,x_t,c)=\log p_\theta(x_0|x_t,c)-\log p_\text{ref}(x_0|x_t,c)$.
> Comparing the loss function above with our loss function in Equation 9, we can see that the core difference lies in the role of $\beta_t$. For (\*), $\beta_t$ is used to regularize the actual reward policy induced by $r(x_0,c)$ while in ours it instead directly controls the policy induced by the implicit reward $r_\theta(x_0,x_t,c,\beta_t)$, which can more effectively shape the model policy $p_\theta$. Furthermore, our formulation also draws a direct connection to a DiffusionDPO-like [1] loss function for discrete diffusion when using $N=2$ samples to estimate the partition $Z_\theta^t$, as we have already demonstrated in Appendix A.4. This reduction also endorses our design while (\*) lacks such an insight.
>
> [1] Wallace et al. Diffusion Model Alignment Using Direct Preference Optimization.
>
> > **Q2. Does the performance of the method degrade if samples from the pretrained model are used instead, for the initial fine-tuning stage?**
>
> Thank you for the good question! We have shown in a new experiment on the DNA task (check the SDPO (w/o Iter) plot in **Fig. 4(a)** of our new manuscript) that the performance does **not** degrade when fine-tuning on samples from the pre-trained reference model. In fact, we trained on a total of ~25k samples generated from our reference model and showed that SDPO was still able to perform strongly, achieving a Pred-HepG2 of 5.0, slightly out-performing our model that was fine-tuned only on the Gosai dataset (without additional data re-labeling).
>
> **We sincerely hope our response could address your concerns!**

---

> ### Author Response · Authors · 2025-11-29
>
> We also performed evaluation of the on-policy baseline diffu-GRPO on the protein inverse folding task:
>
> ||Pred-ddG|%(ddG>0)|scRMSD|%(scRMSD<2)|Success Rate| Entropy|
> |-|-|-|-|-|-|-|
> |diffu-GRPO|1.286|76.8| 1.192|57.1|37.2| 40.0 |
> |SDPO|**1.400**|**87.1**| **0.938**|**88.9**|**75.5**| **42.3** |
>
> Clearly, SDPO yields higher reward while maintaining significantly higher success rate and lower scRMSD, while diffu-GRPO  leads to notable quality degradation. The result again verify the superiority of our approach.
>
>
> **We sincerely hope our response could address your concerns!**

---

### Official Review · Reviewer_WErC · 2025-11-10

**Soundness:** 3
**Presentation:** 3
**Contribution:** 2
**Rating:** 6
**Confidence:** 3

**Summary:**

This paper proposed an offline preference optimization method for aligning the discrete diffusion models with rewards. By decomposing the loss for trajectory alignment into a set of stepwise objectives, the proposed method achieved efficient optimization. Fine-tuned diffusion models outperform RL-based and inference-time guidance methods in terms of reward optimization while maintaining strong sequence quality/naturalness. The efficacy of the method was verified across different domains including DNA, protein, and natural language.

**Strengths:**

1. The paper is well structured and clearly written. The overall presentation flows smoothly and is easy to follow. Most details are clear.
2. The experiments covered diverse relevant domains and are comprehensively evaluated. Beyond achieving the best rewards among all compared methods, the sequences generated by the propose method also maintain comparable naturalness (close to natural sequences, or low NLL) and diversity (entropy).

**Weaknesses:**

1. Missing parts in the related work section. Although the authors did experiments with some inference-time guidance methods in the experiments for DNA and protein, I think it is still worth to mention such baselines in the related work about diffusion alignment. The pros and cons of these two different alignment paradigms, i.e., adding guidance during the inference time without training diffusion models like TDS or CG, or fine-tuning diffusion model through preference optimization like SDPO, could be further clarified in the main text.
2. Although the authors mentioned that dividing a trajectory alignment loss into the objectives for individual steps would make the optimization more efficient, it is not well supported by the experimental results. It could be good if some proxies for training efficiencies, e.g., training speed per batch/convergence speed , are used for comparison between previous preference optimization methods like D2-DPO or DRAKES and SDPO
3. It would be clearer if the complete training algorithm was listed in the paper.

**Questions:**

1. Some clarification questions about methods details.
- Are you directly using CG which involves a time-dependent classifier as proposals in TDS? It seems that in the original TDS paper, they were using the reward model on the clean data $x_0$ which was approximated from $x_t$.
- Is $p_{ref}$ the pertained diffusion model?
- Is each token of $X_0$ a one-hot vector?
- In the first two tasks, DNA and protein, what is the number of Monte-Carlo samples $N$? is $c$ in $r(x,c)$ empty?
2. In the proposed framework, do the reward models always need to be differentiable? From my reading of Eq. 11, differentiability is not necessary.  If not, did you try to apply your method on non-differentiable targets like the docking scores of small molecules?
3. Could you please provide more intuitions about why the ATAC-Acc of the generated DNA sequences decreases but
Pred-Activity increases with a small β, despite their being positively correlated?

---

> ### Author Response · Authors · 2025-11-20
>
> Thank you for the detailed review and suggestions! We provide point-to-point response to your comments as follows.
>
> > **W1. Missing parts in the related work section. It is still worth to mention inference-time guidance methods in the related work about diffusion alignment. The pros and cons of these two different alignment paradigms could be further clarified in the main text.**
>
> Thank you for the suggestion. We have included the following discussions about inference-time guidance approaches in the related work section:
>
> Besides these training-based approaches, inference-time guidance has also been explored for discrete diffusion models to align their sampling distributions. [1] adapts classifier-guidance [2] to discrete diffusion, while sequential Monte Carlo (SMC)-based approaches [3,4,5] have also been introduced for more effective guidance. However, these guidance-based methods usually induce much higher sampling cost and easily suffer from suboptimal performance when the guidance signal is insufficient.
>
> We have also updated the manuscript on openreview correspondingly. Please refer to the related work section in the latest version.
>
> [1] Nisonoff et al. Unlocking guidance for discrete state-space diffusion and flow models.
>
> [2] Ho and Salimans. Classifier-free diffusion guidance.
>
> [3] Wu et al. Practical and asymptotically exact conditional sampling in diffusion models.
>
> [4] Phillips et al. Particle denoising diffusion sampler.
>
> [5] Dou and Song. Diffusion posterior sampling for linear inverse problem solving: A filtering perspective.
>
>
>
> > **W2. Although the authors mentioned that dividing a trajectory alignment loss into the objectives for individual steps would make the optimization more efficient, it is not well supported by the experimental results. It could be good if some proxies for training efficiencies, e.g., training speed per batch/convergence speed, are used for comparison.**
>
> Thank you for the advice! Training efficiency is indeed one of the core advantages of our SDPO, since it does not require extensive online sampling for each training iteration and its loss computation also employs highly efficient MC estimation.
>
> To demonstrate the efficiency of SDPO, we compare SDPO and the baselines on two additional metrics, wallclock time per training step and convergence speed, with results depicted below.
>
> **Time per training step.** We report the wallclock time per each training step of DRAKES, SDPO, and an additional online algorithm diffu-GRPO [1] on the DNA sequence design task:
>
> ||DRAKES|diffu-GRPO|SDPO|
> |-|-|-|-|
> |Time per Step (sec)|6.02|1.51|0.77|
>
> We also compare SDPO to diffu-GRPO on the language modeling task with LLaDA-8B-Instruct:
> ||diffu-GRPO|SDPO|
> |-|-|-|
> |Time per Step (sec)|52.46|21.11|
>
> Notably, without the need to perform online sampling, SDPO is near 7.8$\times$ more efficient than DRAKES and 2.2$\times$ more efficient than diffu-GRPO in terms of wallclock time per training step, while achieving higher rewards and better overall performance, which consistently demonstrates the superiority of our SDPO.
>
> **Convergence speed.** We also perform a systematic head-to-head comparison of the reward convergence speed in **Fig. 4(a)** in the ablation study section of the updated paper. Specifically, our SDPO with iterative labeling achieves 6.2 Pred-HepG2 using only **15k** labeled samples, while DRAKES and diffu-GRPO only achieve 5.6 and 4.2 Pred-HepG2 with **25k** labeled samples.
>
> Putting together, the results verify that SDPO is not only faster at each training step but also much more data-efficient, achieving higher reward with significantly less reward labeling.
>
> We have updated the manuscript to include these training time comparisons and discussions on efficiency. Please kindly refer to the latest version for the detailed updates.
>
> > **W3. It would be clearer if the complete training algorithm was listed in the paper.**
>
> Thank you for the helpful advice! We have revised the manuscript to include the training algorithm as Algorithm 1 in Appendix B.5. Please kindly refer to the updated version.

---

> > ### Author Response · Authors · 2025-11-20
> >
> > > **Q1.1. Are you directly using CG which involves a time-dependent classifier as proposals in TDS? It seems that in the original TDS paper, they were using the reward model on the clean data $x_0$ which was approximated from $x_t$.**
> >
> > For CG we use the reward model over the predicted clean $x_0$ given $x_t$, following the original TDS paper. In general, our baseline implementations strictly follow the established set-up from DRAKES [1]. We also directly borrowed the numbers reported in DRAKES for the guidance-based baselines to ensure credibility.
> >
> > [1] Wang et al. Fine-tuning discrete diffusion models via reward optimization with applications to dna and protein design
> >
> > > **Q1.2. Is $p_\text{ref}$ the pretrained diffusion model?**
> >
> > Yes, $p_\text{ref}$ refers to the pretrained diffusion model.
> >
> >
> > > **Q1.3. Is each token of $\mathbf{x}_0$ a one-hot vector?**
> >
> > Yes. Specifically, $\mathrm{x}_0 \in\mathbb{R}^{n\times m}$ where $n$ is the number of tokens and $m$ is the dimension of the vocabulary. Each token at position $i$ of $\mathbf{x}_0$ is a one-hot vector with dimension $m$.
> >
> > > **Q1.4. In the first two tasks, DNA and protein, what is the number of Monte-Carlo samples $N$? Is $c$ in $r(x,c)$ empty?**
> >
> > * The number of MC samples $N$: For both the DNA and protein experiments, the initial fine-tuning stage uses $N = 25$. In the second and third stages of fine-tuning for the DNA task, $N = 200, 918$, respectively. In the second stage of the protein fine-tuning, $N = 200$.
> > * The condition $c$: In the DNA sequence design task, $c$ is empty (unconditional generation). In the protein inverse folding task, $c$ is the protein 3D structure.
> >
> >
> > > **Q2. In the proposed framework, do the reward models always need to be differentiable? From my reading of Eq. 11, differentiability is not necessary. If not, did you try to apply your method on non-differentiable targets like the docking scores of small molecules?**
> >
> > Our framework never assumes differentiability of the reward model, and SDPO does not backpropagate any gradient through the reward model. Instead, this is indeed one of the key advantages of our SDPO, making it compatible to any sort of reward models.
> >
> > Our experiments are performed on sequence data while computing the docking score of small molecules in addition requires the structure of the molecules, which is not within the scope of the current evaluation.
> > We instead provide additional evaluation in the language modeling task. We adopt the GSM8K dataset with the standard split, where the reward is the binary accuracy (1 or 0) of the model response w.r.t. the questions. In this scenario, our SDPO enhances the accuracy of LLaDA-8B-Instruct from 78.6 to 81.8, which verifies the effectiveness of SDPO under non-differentiable reward.
> >
> >
> > > **Q3. Could you please provide more intuitions about why the ATAC-Acc of the generated DNA sequences decreases but Pred-Activity increases with a small β, despite their being positively correlated?**
> >
> > Yes, Pred-Activity is indeed positively correlated with ATAC-Acc. However, what we see with a smaller $\beta$ value where Pred-Activity increases but ATAC-Acc decreases are signs of reward overfitting: when $\beta$ is too small, the guidance from the reward term in the loss dominates, which potentially leads to overfitting to this stronger reward signal. This is a common trend that occurs for each method, not just SDPO, thus making proper tuning of hyper-parameters essential to achieve a balance between reward optimization and naturalness of samples.
> >
> >
> > **We sincerely hope our response could address your concerns!**

---

### Author Response · Authors · 2025-11-25
**Looking forward to your feedback!**

Dear Reviewers,

Thank you again for your constructive review! We have provided detailed point-to-point response and also performed additional experiments, including e.g. comparisons with on-policy approach, computational efficiency evaluation, and reward correlation analysis.

As the discussion period is coming to the end soon, we are eager to hear back from you about our rebuttal and we are more than willing to offer more explanations if there is still any concern.

Thank you and we look forward to your feedback!

Best,

Authors

---

### Author Response · Authors · 2025-12-02
**Rebuttal Summary**

We sincerely thank all reviewers for the detailed comments. We have provided point-to-point response to all questions raised by the reviewers including additional experiment results. Since we do not hear back from any of the reviewers in the rebuttal phase, we summarize the key points from each reviewer and our response to these concerns.

**Reviewer WErC**:
-  **Efficiency:** We have added a head-to-head comparison of reward convergence rate of our SDPO against DRAKES and diffu-GRPO in Figure 4(a) in the updated manuscript. SDPO demonstrates significantly better efficiency with less training time.
- **Algorithm details:** We have clarified the details in Algorithm 1 in Appendix B.5.

**Reviewer Gnse**:
- **Theoretical correctness:** We have clarified that the posterior $p\_\theta(\mathbf{x}\_0|\mathbf{x}\_t)$ we leveraged throughout the work is exactly the factorized approximation mentioned by the reviewer, and all theoretical derivations  in this paper upon this factorized posterior are correct. For better clarity, we have changed the notation in the updated manuscript into $\hat{p}\_\theta(\mathbf{x}\_0|\mathbf{x}\_t)$ to explicitly highlight the factorization.
- **Comparison against diffu-GRPO:** We have added diffu-GRPO as a baseline in all three experiments, including DNA, protein, and language modeling (see Table 1,2, and 3 in the paper). Our approach consistently outperforms diffu-GRPO by achieving higher reward with better sample naturalness.

**Reviewer pkbk**:
- **Theoretical soundness:** We clarified a potential misconception of the reviewer on the message Theorem 4.1 conveys. In fact, in Theorem 4.1 we present the exact form of the target distribution under SDPO stepwise alignment objective, which explains how SDPO leads to reward enhancement.
- **Comparison against the on-policy approach:** We have added diffu-GRPO as an on-policy baseline in all three experiments, including DNA, protein, and language modeling (see Table 1,2, and 3 in the paper). Our approach consistently outperforms diffu-GRPO by achieving higher reward with better sample naturalness.

**Reviewer KTkz**:
- **DNA evaluation.** We have explained that the baseline DRAKES w/o KL regularization leads to significant reward over-optimization and thus severe degradation in sample quality. We also show that our SDPO can also achieve higher reward than DRAKES w/o KL while maintaining much higher sample naturalness.
- **Additional baselines.** We have added diffu-GRPO and DDPD per the reviewer's suggestion. On all tasks, our method outperforms these baselines by a significant margin.
- **Efficiency and convergence rate.** We have added a head-to-head comparison of reward convergence rate of our SDPO against DRAKES and diffu-GRPO in Figure 4(a) in the updated manuscript. SDPO demonstrates significantly better efficiency with less training time.

Besides these clarifications and additional experiments, we also perform a reward correlation analysis as an in-depth study of our stepwise alignment approach in Figure 4(b) of the updated paper.

Putting together, we have provided a comprehensive response to each point in the review, and we sincerely hope our efforts could be considered and evaluated.

Best,

Authors

---

### Meta-Review · Area_Chair_FWtM · 2026-01-07

**Summary:**

* Theoretical concerns around assuming access to $p_\theta(x_0 | x_t)$ and the strength of conditions needed for Theorem 4.1
* Missing baselines

**Reviewer Concerns:**

Both major concerns were addressed in the rebuttal. The theoretical results and meaning of $p_\theta(x_0 | x_t)$ have been clarified as well as the baselines mentioned by the reviewer have been added.

**Reviewer Scores:**

WErC 6 --> 8: All major concerns were adequately addressed, especially around related work and efficiency.

Gnse 4 -> 6: The main concerns around theory and baselines were addressed

pkbk 4 --> 6: All concerns were addressed especially clarifications around the meaning of theorem 4.1.

KTkz 4 --> 6: Concerns around performance of baselines and additional baselines were addressed

---

### Decision · Program_Chairs · 2026-01-26

Accept (Poster)